# The Repurposed Drugs Suramin and Quinacrine Cooperatively Inhibit SARS-CoV-2 3CL^pro^ In Vitro

**DOI:** 10.3390/v13050873

**Published:** 2021-05-10

**Authors:** Raphael J. Eberle, Danilo S. Olivier, Marcos S. Amaral, Ian Gering, Dieter Willbold, Raghuvir K. Arni, Monika A. Coronado

**Affiliations:** 1Institute of Biological Information Processing (IBI-7: Structural Biochemistry), Forschungszentrum Jülich, 52428 Jülich, Germany; i.gering@fz-juelich.de (I.G.); d.willbold@fz-juelich.de (D.W.); 2Institut für Physikalische Biologie, Heinrich-Heine-Universität Düsseldorf, Universitätsstraße, 40225 Düsseldorf, Germany; 3Campus Cimba, Federal University of Tocantins, Araguaína, TO 77824-838, Brazil; doliviercg@gmail.com; 4Institute of Physics, Federal University of Mato Grosso do Sul, Campo Grande, MS 79070-900, Brazil; marcos.amaral@ufms.br; 5JuStruct: Jülich Centre for Structural Biology, Forchungszentrum Jülich, 52428 Jülich, Germany; 6Multiuser Center for Biomolecular Innovation, IBILCE, Universidade Estadual Paulista (UNESP), São Jose do Rio Preto, SP 15054-000, Brazil; raghuvir.arni@unesp.br

**Keywords:** SARS-CoV-2, COVID-19, 3CL^pro^, main protease, repurposing approved drugs, suramin, quinacrine

## Abstract

Since the first report of a new pneumonia disease in December 2019 (Wuhan, China) the WHO reported more than 148 million confirmed cases and 3.1 million losses globally up to now. The causative agent of COVID-19 (SARS-CoV-2) has spread worldwide, resulting in a pandemic of unprecedented magnitude. To date, several clinically safe and efficient vaccines (e.g., Pfizer-BioNTech, Moderna, Johnson & Johnson, and AstraZeneca COVID-19 vaccines) as well as drugs for emergency use have been approved. However, increasing numbers of SARS-Cov-2 variants make it imminent to identify an alternative way to treat SARS-CoV-2 infections. A well-known strategy to identify molecules with inhibitory potential against SARS-CoV-2 proteins is repurposing clinically developed drugs, e.g., antiparasitic drugs. The results described in this study demonstrated the inhibitory potential of quinacrine and suramin against SARS-CoV-2 main protease (3CL^pro^). Quinacrine and suramin molecules presented a competitive and noncompetitive inhibition mode, respectively, with IC_50_ values in the low micromolar range. Surface plasmon resonance (SPR) experiments demonstrated that quinacrine and suramin alone possessed a moderate or weak affinity with SARS-CoV-2 3CL^pro^ but suramin binding increased quinacrine interaction by around a factor of eight. Using docking and molecular dynamics simulations, we identified a possible binding mode and the amino acids involved in these interactions. Our results suggested that suramin, in combination with quinacrine, showed promising synergistic efficacy to inhibit SARS-CoV-2 3CL^pro^. We suppose that the identification of effective, synergistic drug combinations could lead to the design of better treatments for the COVID-19 disease and repurposable drug candidates offer fast therapeutic breakthroughs, mainly in a pandemic moment.

## 1. Introduction

In December 2019, local health authorities reported an increasing number of pneumonia cases spreading rapidly across the city of Wuhan in the Hubei province in China. The causative agent of this disease was identified as SARS-coronavirus-2 (SARS-CoV-2) [1]. Although most cases are asymptomatic or only evidence mild symptoms with reasonable recovery rates, a small percentage of infected patients develop more severe manifestations, such as severe pneumonia, respiratory failure, multiple organ failures, and numerous cases resulting in death [2]. SARS-CoV-2 has spread worldwide, leading to a coronavirus pandemic of unprecedented magnitude, and the WHO has declared coronavirus disease 2019 (COVID-19) as an international public health emergency [3]. Globally, more than 148 million confirmed cases resulted in over 3.1 million victims, as reported by the WHO on 29^th^ April 2021 [4]. This staggering number is a significant challenge for precarious healthcare systems, especially in developing countries. Under these circumstances, the identification and development of safe and effective SARS-CoV-2 drugs are of high priority.

Coronoviridae forms the largest family of positive-sense single-stranded RNA viruses and is classified into four genera (α, β, γ, and δ) [5]. SARS-CoV, Middle East respiratory syndrome coronavirus (MERS-CoV), and SARS-CoV-2 are β-coronaviruses [6]. The analysis of the SARS-CoV-2 genome sequences revealed a higher identity to bat SARS-like coronavirus (nucleotide similarity: 89.1%) [7]. The genomic RNA of coronaviruses is ~30,000 nucleotides in length and contains at least six open reading frames (ORFs) [8,9]. The first ORF (ORF 1a/b), about two-thirds of the genome length, directly translates two polyproteins, pp1a and pp1ab, and named because there is an a-1 frameshift between ORF1a and ORF1b. The polyproteins are processed by the main protease (M^pro^, also known as 3C-like protease abbreviated as 3CL^pro^) and by the viral papain-like proteases into 16 (in some coronaviruses into 15 due to the absence of nsp1) nonstructural proteins (NSPs). These NSPs are involved in the generation of genomic and subgenomic RNAs that encode four main structural proteins (envelope (E), membrane (M), spike (S), and nucleocapsid (N) proteins) and other accessory proteins [10,11]. Therefore, these proteases, especially 3CL^pro^, play vital roles in the replication process of coronaviruses.

3CL^pro^ is a cysteine protease with three-domain (domains I to III) organization and chymotrypsin-like fold [12]. Active 3CL^pro^ consists of a homodimer containing two protomers, and the coronavirus 3CL^pro^ features a noncanonical Cys-His dyad located in the cleft between domains I and II [12,13,14]. The functional importance of 3CL^pro^ in the viral life cycle combined with the absence of closely related homologous in humans indicates that this protease is an attractive target for the development of antiviral drugs [15]. Thus, so far, three clinically safe and efficient mRNA vaccines against SARS-CoV-2 have been approved (Pfizer-BioNTech, Moderna, Johnson & Johnson, and AstraZeneca Covid 19 vaccines) [16,17,18,19]. Several drugs have shown clinical benefits in certain stages of COVID-19 (e.g., remdesivir, dexamethasone, favipiravir, lopinavir/ritonavir, and darunavir), and some of them have been approved for emergency use [20,21,22,23,24]. However, production limits and speeds in vaccinating the entire worldwide population are gaps that favor the increase in the numbers of SARS-CoV-2 variants, making it imminent to identify alternative drugs that possess the potential to treat the effects caused by COVID-19. An increasing appearance of SARS-CoV-2 variants can be observed worldwide (B.1.1.7, B.1.351, A.23.1, B.1.525, B.1.526, and P.1) [25], and the genomic mutations could lead to the inefficiency of the vaccines.

Thus, repurposing of existing drug molecules could be a rapid alternative to combat the virus infection. The well-described antiparasitic drugs-like chloroquine, quinacrine, and suramin revealed antiviral effects against SARS-Cov-2 in cell cultures [26,27,28,29,30]. These three compounds have been used as antiprotozoal drugs, with chloroquine and quinacrine used against malaria [31,32] and Suramin used against sleeping sickness which is caused by trypanosomes [33].

In the present study, we tested the inhibitory potentials of chloroquine, quinacrine, and suramin against SARS-CoV-2 3CL^pro^. Quinacrine and suramin presented IC_50_ values of <10 µM; however, chloroquine did not affect the SARS-CoV-2 3CL^pro^ activity. Further experiments identified quinacrine and suramin as competitive and noncompetitive inhibitors, respectively. A 1:1 (*v*:*v*) combination of quinacrine and suramin decreased the IC_50_ value to <500 nM.

Fluorescence spectroscopy and surface plasmon resonance (SPR) experiments showed that suramin interaction with SARS-CoV-2 3CL^pro^ induced a conformational change in the protease structure, increasing the quinacrine affinity enormously from around 230 to 30 µM.

Our results let us assume that quinacrine and suramin are attractive drug candidates to combat SARS-Cov-2 infections, and molecular docking and molecular dynamics simulation inferred possible amino acid residues responsible for the interaction between the 3C-like protease and the molecules.

## 2. Materials and Methods

### 2.1. Cloning, Expression, and Purification of SARS-CoV-2 3CL^pro^

The codon-optimized cDNA encoding SARS-CoV-2 3CL^pro^ (Uniprot entry: P0DTD1; virus strain: hCoV-19/Wuhan/WIV04/2019) was synthesized and implemented in the ampicillin-resistant vector pGEX-6P-3 (BioCat GmbH, Heidelberg, Germany). The construct contained an N-terminal GST-tag and a PreScission protease cleavage site (LEFLFQGP). SARS-CoV-2 3CL^pro^-pGEX-6P-3 vectors were transformed into *E. coli* Lemo21 (DE3) (New England BioLabs, Ipswich, MA, USA) competent cells and was grown overnight at 37 °C in an LB medium. This preculture was added to a fresh LB medium (Ampicillin and Chloramphenicol) and grew at 37 °C, until the cells reached an OD_600_ of 0.6. Gene expression was induced with a final concentration of 0.5 mM IPTG (1 mM Rhamnose was added) and incubated for 3 h at 37 °C and 120 rpm. Subsequently, the culture was harvested by centrifugation (4000 rpm) at 5 °C for 20 min (Sorvall RC-5B Plus Superspeed Centrifuge, Thermo Fisher Scientific, Waltham, MA, USA; GSA rotor). The supernatant was discarded. The cells containing the recombinant SARS-CoV-2 3CL^pro^_GST were resuspended in 50 mM Tris-HCl pH 8.0, 200 mM NaCl (lysis buffer) and stored at −20 °C for subsequent purification.

For purification, the cell suspension was incubated on ice for 1 h with the addition of lysozyme; subsequently, it was lysed by sonication in 4 pulses of 30 s each with an amplitude of 30% interspersed by intervals of 10 s. The crude cell extract obtained was centrifuged at 7000 rpm at 6 °C for 90 min. The supernatant containing SARS-CoV-2 3CL^pro^_GST was loaded onto a GSH-Sepharose matrix, which was extensively washed with the lysis buffer. The protein was eluted with the same buffer plus the addition of 10 mM GSH. The eluted fractions were concentrated and dialyzed against PreScission protease cleavage buffer (50 mM Tris (pH: 7.0), 200 mM NaCl, 1 mM DTT, and 1 mM EDTA). PreScission protease was used to cleave the GST-tag from the SARS-CoV-2 3CL^pro^_GST-fused protein. For 100 µg target protein concentration, 10 µg PreScission protease were added, and the sample was incubated at 4 °C for 36 h. The separation of the target protein, the GST-tag, and the PreScission protease was achieved using GSH-Sepharose. Further, to remove aggregated fraction, size exclusion chromatography was used (Superdex 200 10/300 GL GE Healthcare, Chicago, IL, USA), the column was equilibrated with 20 mM Tris-HCL (pH 8.0) and 150 mM NaCl. Sample purity after each purification step was assessed by 15% SDS-PAGE gels. The corresponding protein fraction was concentrated up to 2 mg/mL and stored at –20 °C.

### 2.2. Activity Assay of SARS-CoV-2 3CL^pro^

SARS-CoV-2 3CL^pro^ activity assay was performed as described earlier using a fluorogenic substrate DABCYL-KTSAVLQ↓SGFRKME-EDANS (Bachem, Switzerland) in a buffer containing 20 mM Tris (pH 7.2), 200 mM NaCl, 1 mM EDTA, and 1 mM TCEP [34,35,36]. The reaction mixture was pipetted in a Corning 96-Well plate (Sigma Aldrich) consisting of 0.5 µM protein, and the assay was initiated with the addition of the substrate at a final concentration of 50 µM. The fluorescence intensities were measured at 60 s intervals over 30 min using an Infinite 200 PRO plate reader (Tecan, Männedorf, Switzerland). The temperature was set to 37 °C. The excitation and emission wavelengths were 360 and 460 nm, respectively. For K_M_ and V_max_ measurements, the procedure was followed as described previously [36]. A substrate concentration from 0 to 200 µM was applied. The initial velocity of the proteolytic activity was calculated by linear regression for the first 15 min of the kinetic progress curves. The initial velocity was plotted against the substrate concentration with the classic Michaelis–Menten equation using GraphPad Prism5 software, and K_cat_ was obtained using the Equation (1):K_cat_ = V_max_/[E],(1)
while V_max_ is the experimentally determined maximal velocity and [E] is the enzyme concentration in the experiment [37]. All measurements were performed in triplicate, and data are presented as mean ± SD.

### 2.3. Inhibition Assay of SARS-CoV-2 3CL^pro^

The inhibition of SARS-CoV-2 3CL^pro^ activity by chloroquine (purity: >98%; Sigma-Aldrich, St. Louis, MO, USA), quinacrine (purity: >90%; Sigma-Aldrich, USA), and Suramin (purity: >90%; Sigma-Aldrich, USA) was investigated using the activity assay described above. Compounds with a concentration of 10 µM was used for a preliminary screening test.

Chloroquine did not show a satisfactory inhibitory effect (~10% of inhibition); therefore, it was excluded from the studies. For the final inhibition assays, 0.5 µM of the protein was incubated with 0–100 µM suramin and 0–150 µM quinacrine. The mixtures were incubated at RT for 30 min. When the substrate with a final concentration of 50 µM was added to the mixture, the fluorescence intensities were measured at 60 s intervals over 30 min using an Infinite 200 PRO plate reader (Tecan, Männedorf, Switzerland). The temperature was set to 37 °C, and the excitation and emission wavelengths were 360 and 460 nm, respectively. Inhibition assays were performed as triplicates.

For the quinacrine and suramin combination test, a 1:1 (*v*:*v*) stock solution of the molecules was prepared. The protein (0.5 µM) was incubated with 0–75 µM of the combined molecules (quinacrine and suramin).

The IC_50_ value was calculated by plotting the initial velocity against various concentrations of the combined molecules using a dose–response curve in GraphPad Prism5 software. All measurements were performed in triplicate, and data are presented as mean ± SD.

### 2.4. Assays to Exclude Quinacrine and Suramin as Promiscuous Inhibitors

A detergent-based control was performed to exclude inhibitors, which possibly acted as aggregators of 3CL^pro^ by adding 0.001%, 0.01%, and 0.1% of Triton X-100 to the reaction [38]. Four concentrations of quinacrine (1, 5, 30, and 60 µM), suramin (1, 5, 20, and 60 µM), and quinacrine/suramin combination (0.25, 1, 5, and 25 µM) were tested. All measurements were performed in triplicate, and data are presented as mean ± SD.

We performed colorimetric assay to exclude quinacrine and suramin as compounds that probably induced redox cycling in reducing environments. Hydrogen peroxide (H_2_O_2_) generated by redox cycling compounds incubated with strong reducing agents (e.g., TCEP or DTT) could mediate the oxidation of phenol red (PR) (Sigma-Aldrich, USA) (based on the H_2_O_2_-dependent horseradish peroxidase (HRP; Sigma-Aldrich, USA)-mediated oxidation), which produced a change in its absorbance at 610 nm in alkaline pH [39,40,41]. The assay was performed in Nunc 96-well plates with a flat bottom (Thermofisher Sientific, Waltham, MA, USA), and the final volume was 60 µL. Suramin (0–100 µM) and quinacrine (0–150 µM) were tested in the activity assay buffer as described above, and 1 mM TCEP was added separately. As a plate control, HRP-PR and 100 µM H_2_O_2_ were used. The final HRP-PR detection reagent (100 µg/mL PR and 60 µg/mL HRP) was prepared in Hank’s balanced salt solution (HBSS). HPR-PR and PR without the addition of H_2_O_2_ were used as negative controls. Suramin and quinacrine were incubated with TCEP at RT for 30 min prior to the addition of the HRP-PR detection reagent. After an additional incubation period at RT for 10 min, the assay was terminated by the addition of 10 µL of 1 N NaOH to all wells, and the absorbance of the phenol red was measured at 610 nm using an Infinite 200 PRO plate reader (Tecan, Männedorf, Switzerland). All measurements were performed in triplicate, and data are presented as mean ± SD.

### 2.5. Determination of the Inhibition Mode

The mode of inhibition was determined using different final concentrations of the inhibitors and the substrate. Briefly, SARS-CoV-2 3CL^pro^ at 0.5 µM was incubated with the inhibitor at different concentrations at RT for 30 min. Subsequently, the reaction was initiated by the addition of the corresponding concentration series of the substrate. The data analysis was performed using a Lineweaver–Burk plot; therefore, the reciprocal of velocity (1/V) vs. the reciprocal of the substrate concentration (1/(S)) was compared [42,43]. All measurements were performed in triplicate, and data are presented as mean ± SD.

### 2.6. Determination of the Dissociation Constant Using Surface Plasmon Resonance

The equilibrium dissociation constant (K_D_) of the binding of suramin and quinacrine to SARS-CoV-2 3CL^pro^ was determined by SPR spectroscopy using a Biacore T200 instrument (GE Healthcare, Uppsala, Sweden). 3CL^pro^ was used as the ligand, while suramin and quinacrine were used as analytes. A protease was immobilized onto a series S Carboxymethyl-dextran CM-5 sensor chip (GE Healthcare, Uppsala, Sweden) by amine coupling. The flow cells were activated by a mixture of 50 mM N-hydroxysuccinimide (NHS) and 16.1 mM N-ethyl-N’-(dimethylaminopropyl)carbodiimide (EDC) (XanTec, Düsseldorf, Germany) for 7 min. The protease was diluted to 50 μg/mL in 10 mM sodium acetate (Merck, Darmstadt, Germany) and injected over one of the activated flow cells to a final signal of 3000 resonance units (RU). After the immobilization was completed, the ligand and reference flow cells were deactivated by injecting 1 M ethanolamine (pH: 8.5; XanTec, Düsseldorf, Germany) for 7 min.

For the K_D_ value determination, multicycle experiments were performed with PBS and 0.05% Tween 20 (pH: 7.4; AppliChem, Darmstadt, Germany) as a running buffer at 25 °C and a flow rate of 20 μL/min. The analytes were diluted in the running buffer to the following concentrations, i.e., 0.23, 0.69, 2.06, 6.17, 18.52, 55.56, 166.67, and 500 μM. All samples were injected over the flow cells for 180 s, followed by a dissociation phase of 900 s with the running buffer. The reference flow cell and buffer injections (c = 0 μM) were used to double referencing the sensorgrams. For data evaluation, the sensorgrams were fitted by the steady-state affinity model implemented in the Biacore T200 Evaluation Software 3.2.

For the K_D_ determination of the binding interaction between quinacrine and 3CL^pro^ in the presence of suramin, we performed the following procedures: Quinacrine was diluted in the running buffer to the following concentrations: 0.23, 0.69, 2.06, 6.17, 18.52, and 55.56 µM. All samples and the zero analyte injections, for double referencing, comprised 100 µM suramin. For data processing, the reference flow cell and zero analyte injections (concentration = 0 µM quinacrine + 100 µM Suramin) were used to double reference the sensorgrams. The experiments were performed in triplicate.

### 2.7. Intrinsic Tryptophan (Trp) Fluorescence of SARS-CoV-2 3CL^pro^ under the Influence of Quinacrine and Suramin

The intrinsic Trp fluorescence of SARS-CoV-2 3CL^pro^ was measured under the influence of quinacrine and suramin. Briefly, the protein sample was in the 50 µL mixture of 25 mM Tris-HCl (pH: 8.0) and 150 mM NaCl with a final concentration of 10 μM. The protein solution within the cuvette (path length: 1 cm) was titrated stepwise with a molecule stock solution (0–48 µM) of quinacrine or suramin. To avoid the dilution of the protein, the stock solution of the molecules was treated with the same protein concentration (10 μM protein + 500 µM molecule stock). Measurement was conducted following each titration.

### 2.8. Circular Dichroism (CD) Spectroscopy

CD measurements were carried out with a Jasco J-1100 Spectropolarimeter (Jasco, Germany). Far-UV spectra were measured in 190 to 260 nm using a protein concentration of 4 µM in a 20 mM K_2_HPO_4_/KH_2_PO_4_ solution (pH: 7.5). Cells with a path length of 1 mm were used for the measurements; 15 repeat scans were obtained for each sample, and 5 scans were conducted to establish the respective baselines. The averaged baseline spectrum was subtracted from the averaged sample spectrum. The results are presented as molar ellipticity [θ], according to Equation (2):[θ]_λ_ = θ/(c × 0.001 × l × n),(2)
where θ is the ellipticity measured at the wavelength λ (deg), c is the protein concentration (mol/L), 0.001 is the cell path length (cm), and n is the number of amino acids. The results were analyzed, and the secondary structure content was determined using the CDpro software package [44].

### 2.9. Statistical Analysis

All experiments underwent at least 3 independent repetitions, and all data are expressed as mean ± SD. The statistical significance of the mean values’ differences was assessed with one-way ANOVA, followed by Tukey’s multiple comparison test. Significant differences were considered at *p* < 0.05 (*) and *p* < 0.01 (**). All statistical analyses were performed with GraphPad Prism software version 5 (San Diego, CA, USA).

### 2.10. Systems Information, Molecular Docking, and Ligand Parameterization

The 3D model of SARS-CoV-2 3CL^pro^ was obtained from the PDB database (PDB entry: 6M2Q). The amino acid side chains protonation state was set to pH 7.4 using web-server H++ [45], placed in a water box, neutralized and followed by a 200 ns molecular dynamics (MD) simulation. The representative structure was obtained by clustering analysis. Molecular docking was used to select the best pose for ligand–protein interaction using the Autodock Vina program [46]. A grid box was defined for docking the molecule (quinacrine) near the active site and docking in an proposed allosteric site (suramin) [47,48]. These calculations were performed with the scoring function of the program that ranked several poses for each ligand. The best pose of each ligand was chosen to prepare the 3CL^pro^–quinacrine and –suramin complexes.

Ligands were parameterized using Gaussian 16 [49] at the level of theory HF/6–31G* to optimize their geometry, and their electrostatic potentials were calculated. Restrained electrostatic potential (RESP) charges were determined using Antechamber [50], while general amber force field (GAFF) [51] was used for the missing parameters.

### 2.11. Simulation Setup

All the molecular dynamics simulations were carried out using the Amber18 [52] software package and the FF19SB [53] force field was used to describe the protein atoms interactions, while GAFF and RESP charges described quinacrine and suramin molecules. The systems were solvated in an octahedral box of TIP3P water molecules with at least a 10 Å distance from any solvent atoms between the solutes in each direction, and the system was neutralized when necessary. Initially, energy minimization was performed in two steps to remove low contacts from the initial structures. First, the protein complex was constrained (force constant of 50.0 kcal/mol·Å^2^) and minimized using 5000 steepest descent steps followed by 5000 conjugate gradient steps and 10,000 steps unconstrained energy minimization round. The system was slowly heated from 0 to 298 K for 500 ps under a constant atom number, volume, and temperature (NVT) ensemble, while the protein was restrained with a force constant of 25 kcal/mol·Å^2^. After the heating process, the equilibrium stage was performed using a constant atoms number, pressure, and temperature (NpT) ensemble for 5 ns. Finally, the simulation run was performed for 200 ns and 300 ns for single protein and complexes, respectively, without any restraints, and under the NVT ensemble and, the constant temperature and pressure (1 atm) were controlled by Langevin coupling. Long-range electrostatic interaction was calculated by the particle-mesh Ewald method (PME) [54] with an 8 Å cutoff. The Shake constraints were applied to all bonds involving hydrogen atoms to allow a 2-fs dynamics time step.

### 2.12. Molecular Dynamics Analysis

The CPPTRAJ9 program of AmberTools19 [55] was used to analyze the MD simulations. Root mean square deviation (RMSD) and the radius of gyration (Rg) of Cα were calculated to determine the system quality and stability and to determine the equilibration and convergence of the systems. Protein flexibility was calculated by root mean square fluctuation (RMSF) for all Cα atoms, residue-by-residue over the equilibrated trajectories.

Clustering analysis was performed with the k-means method ranging from 2 to 6, and to access the quality of clustering, the Davies-Bouldin index (DBI) values and silhouette analyses were used.

The interaction energy was calculated using the generalized Born (GB)-Neck2 [56] implicit solvent model (igb = 8). Molecular mechanics/generalized Born surface area (MM/GBSA) energy was computed between the protein and the ligand in a stable regime comprising the last 50 ns of the MD simulation, stripping all the solvents and ions. The web version of POCASA 1.1 was used to determine the active site pocket volume after MD simulations [57].

## 3. Results

### 3.1. Expression and Purification of SARS-CoV-2 3CL^pro^

SARS-CoV-2 3CL^pro^_GST fusion protein was expressed in *E. coli* Lemo21 (DE3) cells and purified using a GSH-Sepharose column (Appendix A). The relevant protein fractions were concentrated and prepared for PreScission protease cleavage to remove the GST-tag. The SDS gel (Appendix A) indicated the cleavage efficiency and the purity of 3CL^pro^. The pure protein was concentrated and applied onto a Superdex 200 10/300 GL size exclusion chromatography (GE Healthcare) to remove aggregated protein species (Appendix A). The CD spectroscopy of 3CL^pro^ indicated that the protein was correctly folded after purification and GST cleavage. The deconvolution of the CD data using the software CDpro [44] showed that the protein secondary structure contained around 28% α-helices and 23% β-strands, which is in agreement with the data from the crystal structure (PDB entry: 6M2Q) that contained 25% α-helices and 28% β-strands (Appendix A).

### 3.2. Activity Assay of SARS-CoV-2 3CL^pro^

The SARS-CoV-2 3CL^pro^ activity was investigated using an assay procedure described earlier [34,35,36], and DABCYL-KTSAVLQ/SGFRKME-EDANS (Bachem, Switzerland) was used as a substrate. Activity assays were performed to obtain the kinetic key values V_max_, K_M_, and K_cat_. A standard curve was generated, converting the relative fluorescence unit (RFU) to the amount of cleaved substrate (µM) (Appendix A). In the next step, the protease enzymatic activity was characterized by measuring V_max_ and K_M_ values; 0.5 µM of the protease was mixed with various concentrations of the substrate (0–200 µM). The initial velocity was measured and plotted against the substrate concentration. Curve fitting with the Michaelis–Menten equation resulted in the best fitting values of K_M_ and V_max_ as 25.47 ± 3.43 µM and 47.52 ± 2.91 µM s^–1^, respectively (Appendix A). The calculated K_cat_/K_M_ ratio was 3731.21 s^–1^ M^–1^, which is similar to the previously reported values of 34,261.1 and 5624 s^–1^ M^–1^ [34,36].

### 3.3. Inhibition Assay of Chloroquine, Quinacrine, and Suramin against SARS-CoV-2 3CL^pro^

A primary inhibition test of the 3 antiparasitic compounds, i.e., chloroquine, quinacrine, and suramin (10 µM), was performed against SARS-CoV-2 3CL^pro^ to screen the best inhibitor against the virus protease (Figure 1).

The primary inhibition tests showed strong effects of quinacrine and suramin against SARS-CoV-2 3CL^pro^ activity. In contrast, chloroquine had a weak effect on 3CL^pro^ proteolytic activity. Suramin and quinacrine possessed an antiviral effect against SARS-CoV-2 in a cell culture [26,27,28,29,30]. As target proteins for quinacrine, ACE2 (SARS-CoV-2 receptor) and furin were identified [28]. Suramin was reported to bind to SARS-CoV-2 nsp12 (RdRp), acting as a potent inhibitor of the RdRp by blocking the binding of RNA to enzymes [58].

Based on the primary inhibition results, quinacrine and suramin were further investigated regarding their inhibitory potential. Our results demonstrated that suramin and quinacrine have inhibitory effects against the 3CL^pro^ of SARS-CoV-2 (Figure 2 and Figure 3, respectively). A detergent-based control was performed to exclude inhibitors that possibly acted as an aggregator by adding 0.001%, 0.01%, and 0.1% Triton X-100 to the reaction. Supposing a molecule would exhibit the significant inhibition of 3CL^pro^, which is diminished by detergent, it is almost certainly acting as an aggregation-based inhibitor [38], which was not observed for quinacrine and suramin (Appendix A). To exclude the capacities of quinacrine and suramin to behave as a redox cycling compounds (RCCs), the H_2_O_2_ assay under the influence of TCEP was performed. RCCs generated micromolar concentrations of H_2_O_2_ in the presence of strong reducing agents (e.g., TCEP), H_2_O_2_ can inhibit the catalytic activity of cysteine proteases, like 3CL^pro^, by oxidizing the active site cysteine [41]. Our results demonstrated that quinacrine and suramin do not produce H_2_O_2_ under the influence of 1 mM TCEP and can be excluded as RCCs (Appendix A).

Quinacrine and suramin inhibited the protease activity of 100% at a concentration of 150 µM (Figure 2B) and at 100 µM (Figure 3B), respectively. Quinacrine had a calculated IC_50_ value of 7.8 ± 0.6 µM (Table 1 and Appendix A). Further experiments identified quinacrine as a competitive inhibitor of 3CL^pro^ (Figure 2C), which means that the molecule interacted directly with amino acid residues located in the active site and/or with the substrate-binding site. In contrast, suramin acted as a noncompetitive inhibitor (Figure 3), showing a type of allosteric inhibition.

The potency of suramin in inhibiting 3CL^pro^ was calculated and demonstrated an IC_50_ value of 6.3 ± 1.4 µM (Table 1 and Appendix A).

The IC_50_ values of both studied molecules are in a similar concentration range to other repurposed drug molecules already published (e.g., boceprevir (4.14 µM) [36], menadione (7.9 µM) [59], and disulfiram (9.35 µM) [60]. Boceprevir is a protease inhibitor used to treat hepatitis caused by the hepatitis C virus (HCV) [61]. Disulfiram is a drug used to support the treatment of chronic alcoholism but has been studied as a possible treatment for cancer [62] and latent HIV infection [63]. Menadione is an intermediate in the chemical synthesis of vitamin K and is allowed as a nutritional supplement and used in the treatment of hypoprothrombinemia [64].

### 3.4. Investigation of SARS-CoV-2 3CL^pro^ Interaction with Suramin and Quinacrine Using Fluorescence Spectroscopy and SPR

To confirm the viability of the assay, we used fluorescence spectroscopy observing the changes in the intrinsic Trp of the SARS-CoV-2 3CL^pro^ following the addition of the inhibitor molecules. The fluorescence changes of Trp can reflect the environmental variation of the protein, and the decreased Trp emission intensity confirmed the complex formation between the protease and the inhibitory molecules. Interestingly, the protein interaction with suramin or quinacrine induced a different excitation shift of the protein tryptophans. Suramin interaction caused a conformational change in the protein structure, which was demonstrated by a red-edge excitation shift (REES) of about 30 nm (330 to 360 nm) (Appendix A). REES is defined by increasing interactions between the fluorophore (Trp) and the surrounding solvent in the ground and excited state [65]. On the contrary, quinacrine interaction induced a weak blue-edge excitation shift (BEES) (Appendix A). Through BEES, the environment hydrophobicity of the Trp residue increased significantly [66].

The dissociation constant (K_D_) for the protease–quinacrine and –suramin interaction was determined using SPR (Appendix A). The experiments were performed in duplicate, and the results are shown as mean ± SD (Table 2 and Appendix A).

Table 2 describes the equilibrium dissociation constant (K_D_) value obtained by SPR to evaluate the binding affinity of suramin (59.7 ± 4.5 µM) and quinacrine (227.9 ± 7.9 µM) in the interaction with SARS-CoV-2 3CL^pro^. The suramin interaction with SARS-CoV-2 3CL^pro^ was around four times stronger compared with the Quinacrine interaction. We assume that this binding involves the sulfonate groups of naphthalene moiety, which are overall six negative charges. Based on the chemical structures of both drugs, chemical modifications could improve the binding properties of the molecules, as described for SARS-CoV-2 3CL^pro^ α-ketoamid inhibitors [34].

The fluorescence experiments let us assume that suramin binding induces a conformational change on the protease. Additional experiments were carried out to understand the modifications caused in the protein after suramin binding, which facilitated the protein interaction with the quinacrine molecule. Therefore, the protease was treated with suramin, and afterward, the K_D_ for the quinacrine interaction was determined. Interestingly, the determined K_D_ value was around eight times lower (30.2 ± 12.7 µM) compared with the value without suramin (227.9 ± 7.9 µM) (Table 2; Appendix A and Appendix A). The fluorescence and SPR experiments let us suggest that the binding of suramin induces conformational changes in SARS-CoV-2 3CL^pro^, which can increase the affinity of protein to quinacrine.

### 3.5. Docking and Molecular Dynamic Simulations of Suramin and Quinacrine with the SARS-CoV-2 3CL^pro^ Structure

The atomic coordinates of SARS-CoV-2 3CL^pro^ (PDB entry: 6M2Q) were used as the initial model. The cluster analysis of the MD simulations demonstrated that cluster #0 showed the best results with the representativeness of over 86.1% of the whole simulation time, and the selected structure appeared during the simulation around 169 ns (Appendix A). SARS-CoV-2 3CL^pro^ is a homodimer with the catalytical dyad His41 and Cys145 [34]. The dimerization of the protease is necessary for catalytic activity; the interaction of each protomer N-terminal Ser1 with Glu166 of the other protomer stabilizes the shape of the S1 pocket, which is essential for the binding of the substrate [12]. Chen et al. suggested that only one protomer of the SARS-CoV 3CL^pro^ dimer is active [67].

Interestingly, after the initial MD simulations of the SARS-CoV-2 3CL^pro^ dimer (Appendix A), one protomer active site collapsed. The pocket volume decreased by around two fold, from 261 to 142 Å3, compared to the other protomer-active sites. The main difference observed between the two protomers succeeding the MD simulation was the interaction between the C-terminal Gln306 and Ile152 of the same protomer. A hydrogen bond was observed between NE2 of Gln306 and the backbone oxygen of Ile152 (Appendix A). We assume that this interaction and the already described interaction of Glu166 with Ser1 stabilize the active site. We carried out theoretical investigations through docking and molecular dynamics simulations to understand the molecular binding modes of the molecules. Quinacrine (competitive inhibitor) was docked in both active sites, and suramin (a highly negatively charged molecule and noncompetitive inhibitor) was docked in a positively charged region on the surface of the protease [68]. Subsequent MD simulations demonstrated a stable system for a period of 300 ns (Appendix A). The binding assessment of the ligands to the protease was observed using a residue-wise decomposition of the MD simulations binding energy. Table 3 and Figure 4 illustrate the 3CL^pro^ amino acid residues contribution to the active sites binding energy that interacted with quinacrine. The interactions between quinacrine and the protease were stabilized by two hydrogen bonds (H-bond), which were mediated by donor and acceptor atoms of Met165 and Gln189 (Table 4). His41, Met49, Val186, Arg188, and Gln194 interacted with quinacrine by hydrophobic interactions (Table 3 and Figure 4).

Figure 5 shows the amino acid residues involved in the interaction between 3CL^pro^ and suramin. Table 3 and Table 4 summarize the H-bonds and hydrophobic interactions. The interactions that stabilized the protease–suramin complex was mediated by six H-bonds that were formed by Lys12, Lys97, Lys100, Tyr101, and Phe103. Additionally, the ligand was stabilized by hydrophobic interactions with the residues, i.e., Lys97, Lys100, Lys102, Phe103, Val104, and Arg105 (Table 3).

As a competitive inhibitor of SARS-CoV-2 3CL^pro^, quinacrine interacted through hydrophobic interaction with His41 of the protease catalytic dyad, blocking the active site entrance.

On the contrary, suramin did not directly interact with the active site residues, and our results let us assume that suramin interacted with the protein allosterically, changing the catalytic site conformation, thus preventing the substrate entry.

An important task is the inhibitors suitability to the binding areas, so that covalent inhibition occurs [69]. Therefore, the nucleophilicity of the amino acid residues in the target and electrophilic groups in the drug needs to be considered [70]. The relative nucleophilicity of the amino acid residues in their neutral states are given in the order of Cys (1) > His (10 − 2) > Met (10 − 3) > Lys, Ser (10 − 5) > Thr and Tyr (10–6) [69,70]. This is one of the reasons why many of the covalent modifiers in drugs are designed to target the thiol group of cysteine [71,72]. The MD results showed that quinacrine interacts directly with nucleophilic residues in the active site, such as His41 and Met165, which makes them highly attractive for covalent interactions. Contrary, suramin interacts with Lys and Tyr residues that can form covalent bonds [73] but with less probability than Cys, His, and Met.

MD analyzes of the 3CL protease complexed to quinacrine or suramin were performed to study its plasticity in more detail according to the substrate-binding sites (S1, S1’, S2, and S3) and the oxyanion hole.

Amino acid residues forming the substrate-binding site of SARS-CoV-2 3CL^pro^ and their subsites were described previously (Appendix A) [73]. Figure 6 illustrates the changes in the SARS-CoV-2 3CL^pro^ substrate-binding site caused by the interaction with the inhibitors. MD simulations revealed a possible mode of interaction of quinacrine with SARS-CoV-2 3CL^pro^. Therefore, His41, one amino acid residue of the catalytic dyad, interacted with quinacrine; this interaction was responsible for partially blocking the S1’ and S3 sites occupying subsite S2 completely (Figure 6). Four residues of the protease S2 subsite interacted directly with quinacrine via hydrophobic interactions (His41, Met49, Asp187, and Arg188), and two residues of the S3 subsite formed H-bonds with the inhibitor molecule (Met165 and Gln189). In the case of suramin, no single amino acid residue of the S1, S1’, S2, and S3 subsites interacted with the ligand, which agrees with the activity assay results describing suramin as a noncompetitive inhibitor of SARS-CoV-2 3CL^pro^.

Suramin binds allosterically to 3CL^pro^. Based on the MD results, we suggest that the ligand interaction induces conformational changes in the catalytic site (most observable for subsites S1’ and S2), thus preventing the entry and the turnover of the substrate (Figure 7). Additionally, the volume of the active site pocket decreased considerably after suramin binding, even whether the molecule was docking just in protomer 1, changing the volume of the pocket from 261 to 90 Å^3^ (protomer 1) and to 120 (protomer 2) (Figure 7C–E).

### 3.6. Suramin and Quinacrine Act Cooperatively to Inhibit SARS-CoV-2 3CL^pro^

Suramin and quinacrine, as described previously, bound in different regions of SARS-CoV-2 3CL^pro^. To conjecture the effects of both molecules simultaneously, we performed a combined inhibitory assay, where both molecules were mixed (volume ratio: 1:1) and tested against SARS-CoV-2 3CL^pro^ (Figure 8). The inhibition assay was performed with a solution containing both ligands (suramin and quinacrine) with a molar ratio of 1:1. The calculated IC_50_ value (0.46 ± 0.1 µM) revealed an increased inhibition capacity of the drugs (Appendix A), and when compared with the single molecules, the combination enhanced (approximately 10 times) the inhibitory capacity of the studied drugs.

Additionally, the docking of quinacrine in each active site of the SARS-CoV-2 3CL^pro^ complex with suramin was performed, and subsequent MD simulations demonstrated a stable system over 200 ns (Appendix A). As described before, one active site of the 3CL^pro^ dimer collapsed, and quinacrine left the active site. In contrast, the second quinacrine stayed in the active site, but the position changed and the amino acids involved in the interaction changed. However, His41 was still involved in this interaction network (Figure 8C,D). This MD simulations experiment let us assume that part of suramin interactions with Lys102, Phe103, Val104, and Arg105 move away, but two H-bonds could be observed between Lys12 and Thr98. Interestingly, Lys97 of the other monomer formed an H-bond, which was not observed before (Figure 8E,F). The differences in the 3CL^pro^ residues interacting with quinacrine, suramin, and quinacrine + suramin are presented in Appendix A.

The performed fluorescence spectroscopy and SPR experiments showed that suramin interaction with SARS-CoV-2 3CL^pro^ increased the affinity with quinacrine dramatically. The results of the MD simulations described above showed a possible interaction mode and let us assume that conformational changes after suramin interaction with SARS-CoV-2 3CL^pro^ could induce a structural change in one active site in the dimer, which made this area more accessible for the interaction with quinacrine (Figure 8).

## 4. Conclusions

Repurposing of clinically developed drugs is a common procedure to combat fast-evolving pathogens. Similar studies addressed the use of established drugs against SARS-CoV-2 (e.g., boceprevir, menadione, and disulfiram) (Table 5) [36,59,60]. The 1:1 (*v*:*v*) combination of quinacrine and suramin possessed an effective anti-3CL^pro^ activity, demonstrating the potential of repurposing drugs to stop the replication process of SARS-CoV-2. However, in vitro cell culture experiments and in vivo experiments are needed to validate that quinacrine and suramin cooperatively inhibit 3CL^pro^ and could lead to a possible alternative to combat COVID-19 infections.

The MD simulations of quinacrine, suramin, and quinacrine + suramin in complexes with the protease demonstrated the potential binding regions on the protein; however, experimental methods (e.g., mutagenesis) are necessary to confirm these observations.

Several variants of SARS-CoV-2 have been identified, namely B.1.1.7, B.1.351, A.23.1, B.1.525, B.1.526, and P.1 [25]. The community transmission of the virus and antiviral treatments can engender novel mutations in the virus, potentially resulting in more virulent strains with higher mortality rates or the emergence of strains resistant to treatment and/or vaccines [74,75]. Mutations were observed in several genes (e.g., *Orf1ab*, *Orf3*, *Orf8*, and envelope protein), but with substitution in the gene encoding, the spike glycoprotein was highly observed (Appendix A). Over several months, the D614G mutation has become the dominant form of the virus circulating globally, with the major mutations occurring on Spike protein. However, in the animal model, it was shown that the variant G614 was not more pathogenic than the original D614 [76]. The analysis of the evolving geographical diversity in SARS-CoV-2 exhibited high similarities and no variation in the sequences of 3CL^pro^ [77]. Potential drugs targeting this enzyme could help to combat the different variants.

The SPR experiments showed that suramin and quinacrine alone possessed a moderate or weak affinity for SARS-CoV-2 3CL^pro^. However, the treatment of the protease with suramin increased the affinity to quinacrine and to the protease by around the factor of eight (230 to 30 µM). We suppose that conformational changes induced by the binding of suramin make an active site more accessible for quinacrine interaction.

## Figures and Tables

**Figure 1 viruses-13-00873-f001:**
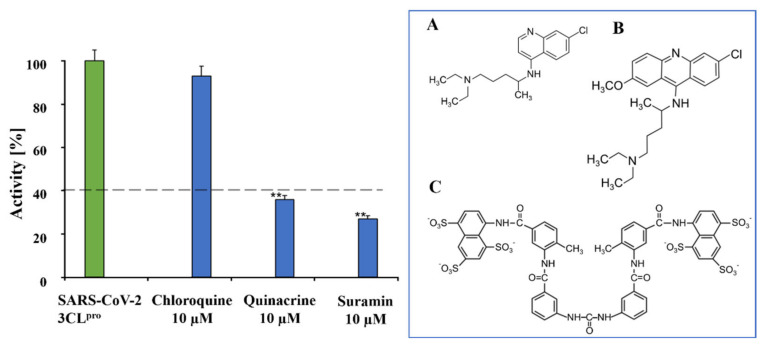
Preliminary inhibition tests of chloroquine, quinacrine, and suramin against SARS-CoV-2 3CL^pro^. Quinacrine and suramin inhibited the virus protease activity by more than 60%. On the contrary, chloroquine showed no relevant inhibition, just about 10%. The blue box shows the molecular structures of chloroquine (**A**), quinacrine (**B**), and suramin (**C**). Data shown are mean ± SD from 3 independent measurements (*n* = 3). Asterisks mean that the data differ from the control (0 µM inhibitor) significantly at *p* < 0.01 (**) according to ANOVA and Tukey’s test.

**Figure 2 viruses-13-00873-f002:**
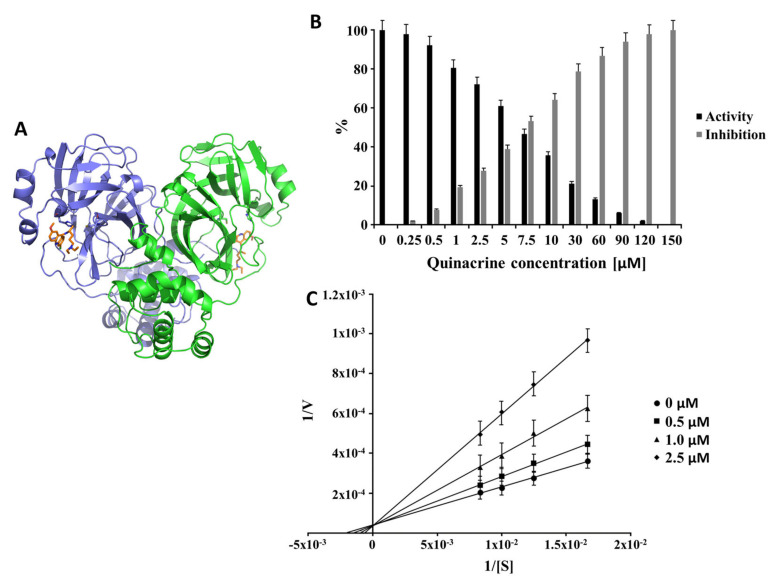
Inhibition effect and inhibition mode of quinacrine over SARS-CoV-2 3CL^pro^: (**A**) 3D representation of the 3CL^pro^-quinacrine complex (3CL^pro^ are indicated by ribbon presentation, and quinacrine is represented by sticks). 3CL^pro^ monomers are colored in green and blue, and quinacrine is colored in orange. The normalized activity and inhibition of the virus proteases, as well as Lineweaver-Burk plots to determine the inhibition mode, are presented. [S] is the substrate concentration; v is the initial reaction rate; (**B**) normalized activity and inhibition of SARS-CoV-2 3CL^pro^ under the quinacrine influence. (**C**) Lineweaver-Burk plots for the quinacrine inhibition of SARS-CoV-2 3CL^pro^. Data shown are mean ± SD from 3 independent measurements (*n* = 3).

**Figure 3 viruses-13-00873-f003:**
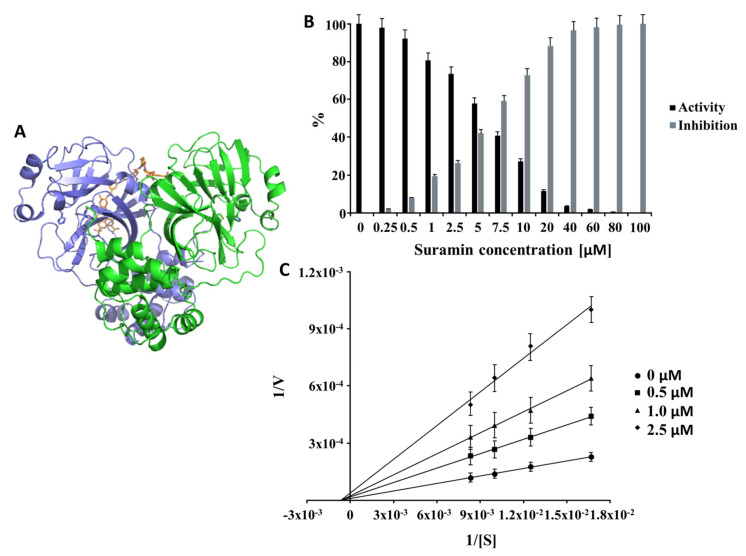
Inhibition effect and inhibition mode of suramin over SARS-CoV-2 3CL^pro^. (**A**) 3D representation of the 3CL^pro^-suramin complex (3CL^pro^ are in ribbon presentation, and suramin is indicated by sticks). 3CL^pro^ monomers are colored in green and blue and suramin in orange. Normalized activity and inhibition of the virus proteases and Lineweaver­-Burk plots to determine the inhibition mode are presented. [S] is the substrate concentration; v is the initial reaction rate. (**B**) Normalized activity and inhibition of SARS-CoV-2 3CL^pro^ under suramin influence. (**C**) Lineweaver-Burk plot for Suramin inhibition of SARS-CoV-2 3CL^pro^. Data shown are mean ± SD from 3 independent measurements (*n* = 3).

**Figure 4 viruses-13-00873-f004:**
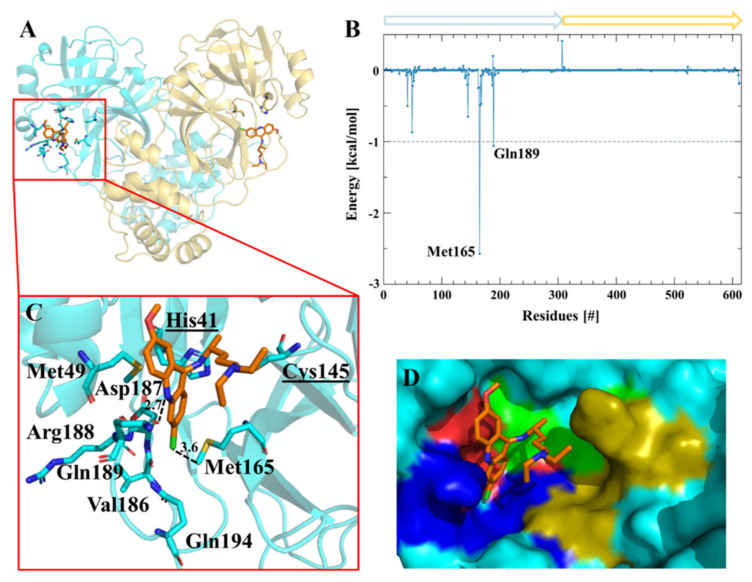
Amino acids participating in the SARS-CoV-2 3CL^pro^-quinacrine interaction. (**A**) Ribbon and sticks representation of the SARS-CoV-2 3CL^pro^-quinacrine complex after molecular dynamics (MD) simulation. (**B**) Decomposition of the binding energy of SARS-CoV-2 3CL^pro^-quinacrine complex. Arrows label protomer A (turquoise) and protomer B (gold). (**C**) SARS-CoV-2 3CL^pro^ amino acids involved in the interaction with quinacrine based on MD simulations. (**D**) Quinacrine interaction in the protease substrate-binding and active site. The substrate-binding subsites are highlighted in green for S1’, gold for S1, red for S2, and blue for S3.

**Figure 5 viruses-13-00873-f005:**
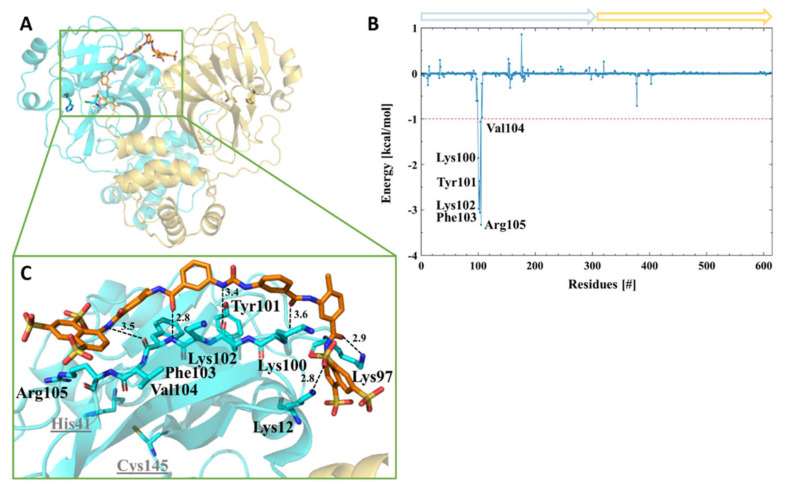
Interface interaction of the SARS-CoV-2 3CL^pro^-suramin complex. (**A**) Ribbon and sticks representation of the SARS-CoV-2 3CL^pro^-suramin complex after MD simulation. (**B**) Decomposition of the binding energy of the SARS-CoV-2 3CL^pro^-suramin complex. Arrows label protomer A (turquoise) and protomer B (gold). (**C**) SARS-CoV-2 3CL^pro^ amino acids involved in the interaction with suramin based on MD simulations.

**Figure 6 viruses-13-00873-f006:**
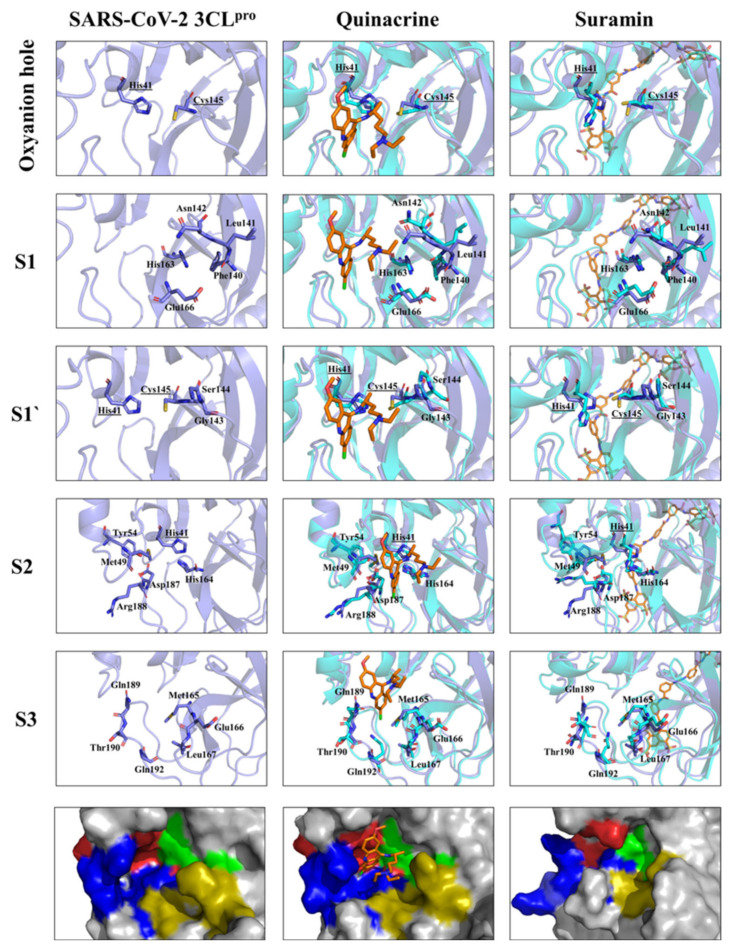
Substrate-binding pocket and oxyanion hole of SARS-CoV-2 3CL^pro^ (S1, S1’, S2, and S3) free accessibility and under quinacrine and suramin influence. The structural overlay between the single protein and in complex, the active site residues are highlighted. The amino acid residues are shown in sticks. The surface view (zoom) of the substrate-binding area demonstrates the occupied area by the molecules and the conformational changes induced by its binding. Substrate-binding subsites highlighted in green for S1’, gold for S1, red for S2, and blue for S3.

**Figure 7 viruses-13-00873-f007:**
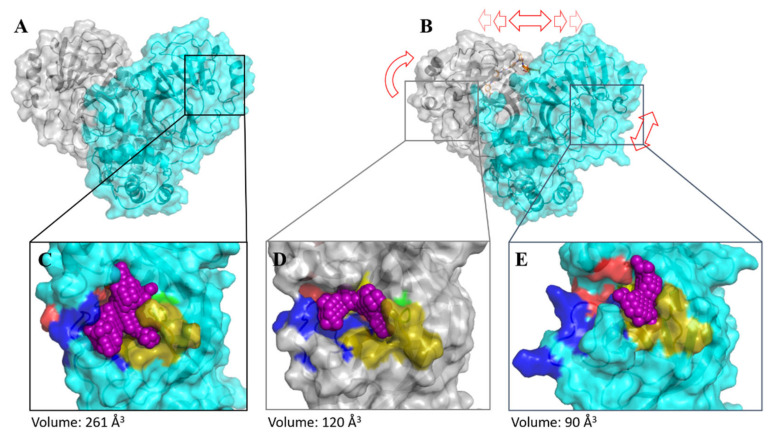
Comparison of SARS-CoV-2 3CL^pro^ with and without suramin after MD simulation. Protomer 1 is colored in turquoise, protomer 2 is colored in grey, and the active site pocket volume is colored in violet. Red arrows highlight a conformational change in the SARS-CoV-2 3CL^pro^–suramin complex. (**A**) SARS-CoV-2 3CL^pro^ dimer. (**B**) SARS-CoV-2 3CL^pro^–suramin complex. (**C**) Active site pocket volume of SARS-CoV-2 3CL^pro^. Substrate-binding subsites are highlighted in green for S1, gold for S1, red for S2, and blue for S3. (**D**) Active site pocket volume of protomer 2. (**E**) Active site pocket volume of protomer 1.

**Figure 8 viruses-13-00873-f008:**
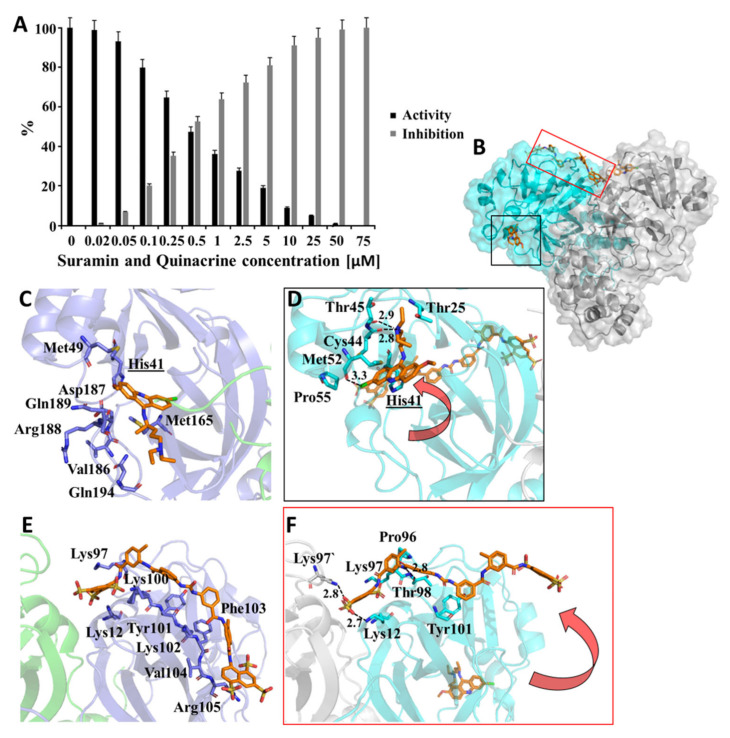
SARS-CoV-2 3CL^pro^ inhibition by a combination of quinacrine and suramin. (**A**) Normalized activity and inhibition of SARS-CoV-2 3CL^pro^ under quinacrine and suramin influence. Data shown are mean ± SD from 3 independent measurements (*n* = 3). (**B**) Ribbon and surface representation of the SARS-CoV-2 3CL^pro^ model (grey/cyan) (PDB entry: 6M2Q) in complexes with suramin and quinacrine (sticks in orange). (**C**) SARS-CoV-2 3CL^pro^ amino acids involved in the interaction with quinacrine. (**D**) SARS-CoV-2 3CL^pro^ amino acids involved in the interaction with quinacrine under the influence of a simultaneous suramin interaction. (**E**) SARS-CoV-2 3CL^pro^ amino acids involved in the interaction with suramin. (**F**) SARS-CoV-2 3CL^pro^ amino acids involved in the interaction with suramin under the influence of a simultaneous quinacrine interaction. (**C**–**F**) are based on MD simulations.

**Table 1 viruses-13-00873-t001:** Summary of the SARS-CoV-2 3CL^pro^ inhibition experiments by quinacrine and suramin.

Molecule	IC_50_ (µM)	Inhibition Type
Quinacrine	7.8 ± 0.6	Competitive
Suramin	6.3 ± 1.4	Noncompetitive

**Table 2 viruses-13-00873-t002:** Summary of the SARS-CoV-2 3CL^pro^ in the complexes with quinacrine and suramin.

Molecule	K_D_ (µM)	Excitation Shift
Quinacrine	227.9 ± 7.9	Blue-edge excitation shift
Suramin	59.7 ± 4.5	Red-edge excitation shift
Quinacrine *	30.2 ± 12.7	-

* K_D_ determination of quinacrine to SARS-CoV-2 3CL^pro^ in the presence of suramin.

**Table 3 viruses-13-00873-t003:** SARS-CoV-2 3CL^pro^ residues involved in forming H-bonds and hydrophobic contacts with quinacrine and suramin.

Ligand	Interacting Residues	
	H-Bond	Hydrophobic interaction
Quinacrine	Met165 and Gln189	His41, Met49, Val186, Asp187, Arg188, and Gln194
Suramin	Lys12, Lys97, Lys100, Tyr101, and Phe103	Lys97, Lys100, Lys102, Phe103, Val104, and Arg105

**Table 4 viruses-13-00873-t004:** Atoms involved in the H-bond interaction between SARS-CoV-2 3CL^pro^ and inhibitors.

Ligand	Residue	Atom (Ligand) *	H-Bond Donor/Acceptor	Distance (Å)
Quinacrine	Met165			
	(side chain) CE	CL1	**C–H---CL**	3.6
	Gln189			
	(side chain) OE1	N3	**O---H–N**	2.7
Suramin	Lys12			
	(side chain) NZ	O18	**N–H---O**	2.8
	Lys97			
	(Side chain) NZ	O14	**N–H---O**	2.9
	Lys100			
	(Backbone) N	O13	**N–H---O**	3.6
	Tyr101			
	(Backbone) O	N3	**O---H–N**	3.4
	Phe103			
	(Backbone) N	O11	**N–H---O**	2.8
	(Backbone) O	N1	**O---H–N**	3.5

* Atom numbers of quinacrine and suramin are shown in Appendix A.

**Table 5 viruses-13-00873-t005:** Examples of drugs or preclinically tested compounds affecting SARS-CoV-2 3CL^pro^.

Compound	IC_50_ (µM)	Inhibition Type ^1^	Reference
Quinacrine	7.8 ± 0.6	Competitive	Reported here
Suramin	6.3 ± 1.4	Noncompetitive	Reported here
Quinacrine + suramin	0.46 ± 0.1	Competitive + Noncompetitive	Reported here
Boceprevir	4.14 ± 0.61	Competitive	[36]
Narlaprevir	5.73 ± 0.67	Competitive	[36]
Simeprevir	13.74 ± 3.75	Competitive	[36]
Auranofin	0.51 ^2^	Competitive	[59]
Menadion	7.96 ^2^	Competitive	[59]
Ebselen	0.67 ± 0.06	Competitive	[60]
Disulfiram	9.35 ± 0.18	Competitive	[60]
Tideglusib	1.55 ± 0.30	Competitive	[60]
Carmofur	1.82 ± 0.06	Competitive	[60]

^1^ Competitive inhibitor binds in an active site or substrate-binding site. The noncompetitive inhibitor binds in an allosteric site. ^2^ Results from He et al. without standard deviation [59].

## Data Availability

All data are reported in the text and Appendix A.

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
