# Peer review of "The Repurposed Drugs Suramin and Quinacrine Cooperatively Inhibit SARS-CoV-2 3CLpro In Vitro"

_viruses, 2021, doi:10.3390/v13050873_

Round 1

Reviewer 1 Report

publish as is.

Author Response

The reviewer suggested "publish as is". No questions and suggestions were send, we didn´t send a response. Thank you to reviewer for his positive feedback.

Reviewer 2 Report

This study focus on repurposed drugs against SARS-CoV-2. The authors found two drugs quinacrine and suramin could inhibit SARS-CoV-2 3CLpro activity competitive and non-competitive, respectively. Different biomedical experiments were carried out to prove the binding between the drugs with SARS-CoV-2 3CLpro. And the two drugs were further proved to work synergistically by inhibiting SARS-CoV-2 3CLpro activity. Recent study from other groups indicate that suramin inhibits SARS-CoV-2 by targeting its RdRp, and quinacrine may target ACE2 and furin to inhibit SARS-CoV-2. So it’s kind of interest that the authors here declare that quinacrine and suramin could also bind to SARS-CoV-2 3CLpro and inhibit its activity.

Recommendations and suggestions

  1. The experiments were well designed and fully supported the conclusion that quinacrine and suramin could bind SARS-CoV-2 3CLproand inhibit its activity. However, the study lack of validation in in vitro cell models. Although, there are published data show that quinacrine and suramin could inhibit SARS-CoV-2 replication in cell models, the inhibiting effects were due to the drugs binding to RdRp and ACE2/furin, respectively, rather than 3CLpro. Thus, I am wondering how the authors could prove that quinacrine and suramin can inhibit SARS-CoV-2 replication in cell models by impairing 3CLpro activity.

  1. The two drugs have already been reported to inhibit SARS-CoV-2 replication by targeting RdRp and ACE2/furin, respectively. So I’m curious about how the authors could come up with the idea to test their effects on 3CLpro.

  1. The authors predicted the possible binding mode of the two drugs against 3CLpro, and also the key residues involved in the binding. However, these results are speculative, and I would recommend further mutagenesis study which will make the conclusion more convincing.

  1. Line21. Johnson & Johnson vaccine are not included.

  1. Line 76. Same as above.

  1. Line 336. 3CLpro should be replaced by 3CLpro.

Author Response

Reviewer 2

Comment 1
The experiments were well designed and fully supported the conclusion that Quinacrine and Suramin could bind SARS-CoV-2 3CLproand inhibit its activity. However, the study lack of validation in in vitro cell models. Although, there are published data show that Quinacrine and Suramin could inhibit SARS-CoV-2 replication in cell models, the inhibiting effects were due to the drugs binding to RdRp and ACE2/furin, respectively, rather than 3CLpro. Thus, I am wondering how the authors could prove that Quinacrine and Suramin can inhibit SARS-CoV-2 replication in cell models by impairing 3CLproactivity.

Response 1
Dear Referee, we agree with you that with our results, we cannot confirm the inhibition of the virus replication, just the inhibition of the protease in vitro. However, according to our results, we demonstrated with strong evidence that it could happen. Of course, to confirm our finding against the whole virus complexity we have to perform a cell-based antiviral assay, nevertheless, using subgenomic replicons, which enable uncoupling of viral replication genes, in our case, 3CL protease. These experiments will allow us to assess the efficacy and bioavailability of drug candidates. Whether we perform the experiment propagating the whole virus, we could not confirm that the inhibition directly affects the protease. As we already informed in the previous revision, we would like to perform cell-based assays. Still, we could not find a collaboration that could perform cell-based assay using subgenomic replicons.
In this manuscript, we are focusing on identifying repurposed drugs that can be used against SARS-CoV-2 3CLproor as a lead molecule. According to our drive, we revealed that Suramin and Quinacrine, using different methodologies, are “promising” available drugs to be used against SARS-CoV-2 infections. The publication of this study will bring relevant information to the scientific community and the availability of the obtained results. These drugs can be tested in cell culture or animal models by another group interested in the results that we presented here.
We included in the Conclusion section the following sentence: However, in vitro cell culture experiments and in vivo experiments are needed to validate that Quinacrine and Suramin cooperatively inhibit 3CLpro and could lead to a possible alternative to combat COVID-19 infections (lines 622-624).
More and more variants of SARS-CoV-2 appear (e.g., B.1.1.7, B.1.351, and P.1), carrying mutations in several genes, e.g., Orf1, Orf3, Orf8, envelope protein, N-terminus of the polyprotein, but especially in the spike glycoprotein. However, the Indian double mutant B.1.617 carry a mutation in the RdRp gene,P4715L (DOI:10.1101/2021.04.23.441101), which can also be observed in B.1.526 (see Supplementary Table S4). The possibility is given that may arise further mutations in the RdRp gene that reduces the binding and inhibition by Suramin and Quinacrine. In that case, it would be of interest to have a second virus target protein inhibited by Quinacrine and Suramin, e.g. 3CLpro. So far, the 3CL protease in the identified variants is unaffected from the mutations, and potential inhibitors targeting the protease could help combat different variants.
Quinacrine and Suramin also affect the Spike/furin/ACE2/ system, but it exists evidence that SARS-CoV-2 can use alternative receptors to enter the cells, e.g. Integrins (DOI: 10.1016/j.antiviral.2020.104759; DOI: 10.3390/v13020146; DOI: 10.3390/v13040645).Thereby, Integrins may function as an alternative receptor for SARS-CoV-2, and the virus could so avoid the entrance by the inhibited ACE2/furin system.
The description of the inhibitory potential of Quinacrine and Suramin against SARS-CoV-2 3CLprooffers the potential to introduce a new target protein for both molecules. Especially with the background of upcoming variants or alternative entrance strategies of the virus.

Comment 2
The two drugs have already been reported to inhibit SARS-CoV-2 replication by targeting RdRp and ACE2/furin, respectively. So I’m curious about how the authors could come up with the idea to test their effects on 3CLpro.

Response 2
We had in previous studies very good inhibition results (in vitro) of Quinacrine and Suramin against flavivirus proteases (ZIKV, DENV, YFV and WNV NS2B/NS3pro) and alphavirus proteases (CHIKV nsP2pro). The suramin effect against ZIKV NS2B/NS3prowas already published (DOI: 10.1016/j.antiviral.2018.10.019). We decided, therefore, to test both drugs against SARS-CoV-2 3CLproactivity. As mentioned in response 1, our results identified SARS-CoV-2 3CLproas an alternative target for Quinacrine and Suramin.

Comment 3
The authors predicted the possible binding mode of the two drugs against 3CLpro, and also the key residues involved in the binding. However, these results are speculative, and I would recommend further mutagenesis study which will make the conclusion more convincing.

Response 3
As described previously, we are focusing on identifying repurposed drugs that can be used against SARS-CoV-2 3CLproor as a lead molecule by combining different methods to characterize and describe the inhibition mode of two drug molecules. We suggested possible binding regions using MD simulations through speculative ideas, yes, it is hypothetical, but it is the way basic science works with the initial questions. We make clear in the manuscript that the binding regions we are showing have the potential to bind the ligands. Still, experimental methods (e.g. mutagenesis) are necessary to confirm the observations (Conclusion lines 625-627).
The MD simulation data described in this manuscript agrees with our experimental data. In the case of Suramin, the allosteric site is in a region described previously:
1.Sencanski et al. 2020 (DOI: 10.3390/molecules25173830)
2.Carli et al. 2021 (DOI: 10.1021/acs.jpclett.0c03182)
3.Günther et al. 2021 (Compound: AT7519, partially) (DOI: 10.1126/science.abf7945)
We identified Quinacrine as a competitive inhibitor and Suramin as a noncompetitive inhibitor. As a competitive inhibitor, Quinacrine competes with the substrate binding in a specific site (active or substrate-binding sites). Our MD simulations results show that the Quinacrine/3CLprocomplex is stable over 200 ns. The same was observed for Suramin, a total period of over 200 ns of MD simulations (details in the section “simulation setup”), which is longer than other published simulations in the literature (examples, see below):
200 ns
https://doi.org/10.1038/s41598-019-42935-y
150 ns
https://doi.org/10.1080/07391102.2015.1046934https://doi.org/10.1016/j.jmgm.2017.03.002
100 ns
https://doi.org/10.1016/j.lfs.2020.118080
50 ns
https://doi.org/10.1080/07391102.2020.1808077

Comment 4
Line21. Johnson & Johnson vaccine are not included.
Response 4
The Johnson & Johnson vaccine was included accordingly in line 22.

Comment 5
Line 76. Same as above.
Response 5
The Johnson & Johnson vaccine and the corresponding reference was included accordingly in lines 78 and 79.

Comment 6
Line 336. 3CLproshould be replaced by 3CLpro.
Response 6
3CLprowas corrected in line 336.

Reviewer 3 Report

In this work, the authors present the in silico drug repurposing of Suramin and Quinacrine as SARS-CoV-2 main protease inhibitors, along with in vitro data confirming their hypothesis. The authors found that both drugs weakly inhibit Mpro, but the level of inhibition increases when applied cooperatively. Quinacrine presents a competitive, while Suramin noncompetitive inhibition mode. 
Regarding the computational part of the study, it is designed properly, using modern state-of-art methods. The molecular dynamics simulations are presented and analysed in detail. The in silico and in vitro results are mutually supported. 
However, I don't feel enough qualified to judge the language quality and experimental procedure design. Anyway, I would recommend acceptance of this paper in its present form.

Author Response

The reviewer recommend acceptance of this paper in the present form. No questions and suggestions were send, we didn´t send a response. Thank you to reviewer for his positive feedback.

Round 2

Reviewer 2 Report

The authors addressed all my questions, no more concerns or suggestions. I recommend acceptance of the paper.

This manuscript is a resubmission of an earlier submission. The following is a list of the peer review reports and author responses from that submission.

Round 1

Reviewer 1 Report

This study titled “The repurposed drugs suramin and quinacrine inhibit cooperatively in vitro SARS-CoV-2 3CLpro” by Eberle et al is of important findings to aid in the treatment against SARS-CoV-2 when no effective therapeutic treatments have yet been discovered. This study evaluates the ability of the repurposing of two clinically approved drugs, suramin and quinacrine to treat SARS-CoV-2.

This study demonstrates strong preliminary support for trying these drugs to treat SARS-CoV-2, however, a proof-of-concept showing that they are actually effective in vitro in the context of a virus infection would be important.

Major corrections:

  • Since the authors conclude that the combination of quinacrine and suramin have the potential to stop the replication process of SARS-CoV-2, further experiments should be performed to strengthen this conclusion. Specifically, in vitro cell culture experiments should be performed, in cell lines such as Vero cells or Calu3 cells. These experiments could be individual assessments of the repurposed drugs, or synergy assays.
  • Since suramin and quinacrine target specific amino acids on the SARS-CoV-2 3CLpro, one concern that may arise is mutations that cause SARS-CoV-2 to escape their effects. In this manuscript, there is no mention of the specific strain that was used to perform these experiments. Do all strains possess these same amino acids at 3CLpro? Please elaborate on which strain that was used. In addition, if these amino acids differ, to strengthen the argument please elaborate on the potential efficacy of suramin and quinacrine on different SARS-CoV-2 variants (ex. The UK variant B.1.1.7, The South African variant, or other variants that may arise in the future).

Minor corrections:

  • The title should be changed to “The repurposed drugs suramin and quinacrine cooperatively inhibit SARS-CoV-2 3CLpro in vitro
  • The use of articles (ex. ‘the, an’) is lacking in some parts of the manuscript. Please proofread and add as necessary.
  • The sentence “Drug repositioning offers hope to the SARS-CoV-2 control” at lines 31-32 does not make sense. Please modify this sentence to improve the meaning such as “Drug repositioning offers hope to control the SARS-CoV-2 pandemic.”
  • At line 71, the author states that there are currently no efficacious drugs and vaccines available to combat SARS-CoV-2 infection, however, there are now authorized vaccines for use in humans (ex. Moderna, Pfizer-BioNTech COVID-19 vaccine). This sentence should address these authorized vaccines, their potential drawbacks, and how drug treatments such as suramin and quinacrine can fill these disadvantages.
  • At line 108, the sentence ‘which was previously extensively washed with the lysis buffer and was extensively washed with the same buffer’ is unclear. Please change this sentence to be grammatically correct and clearer to the reader.
  • Please change ‘has grown overnight’ to ‘was grown overnight’ at line 95.
  • At lines 145-147, the authors state that chloroquine did not show a satisfactory inhibitory effect. Please elaborate on this by including the range of concentrations tested in this experiment.
  • Please remove the extra period at line 192.
  • Throughout the manuscript the use of numerical numbers below ten are used (ex. 1-10), however, in line 198 the word ‘four’ is used. Please make sure the numbers are consistent (ex. Change to 4).
  • At line 275 “Which is in good agreements...” is not a complete sentence. Please improve this sentence by changing the wording or adding it to the previous sentence.
  • From lines 334-337, the authors mention other repurposed drug molecules (boceprevir, menadione, and disulfiram). Please briefly describe what these drugs do.
  • At line 485, change ‘approx.’ to approximately.
  • At line 299, the link to access the supplementary materials does not work.

Author Response

Comment1

Since the authors conclude that the combination of Quinacrine and Suramin have the potential to stop the replication process of SARS-CoV-2, further experiments should be performed to strengthen this conclusion. Specifically, in vitro cell culture experiments should be performed, in cell lines such as Vero cells or Calu3 cells. These experiments could be individual assessments of the repurposed drugs, or synergy assays.

Response 1That’s true and we were trying and still try to find partners that could perform experiments with viral replicons for us,to investigate the effect of the compounds on the SARS-CoV-2 replication process.Unfortunately, we could not find a collaborator partner in time to perform in vitro cell culture experiments. However, we are still looking for partners. We changed our statements and conclusion accordingly in the manuscript.We are assuming that these drug combinations could be a possible inhibitor against the SARS-CoV-2 replication process,perhaps some readers could be interested in performing the in vitro cell tests.However, our results demonstrate that both molecules can be used as lead compounds to develop specific covid-19 inhibitors, which could be based on the molecule structures of Quinacrine and Suramin. It is already very well characterized that the identification of a lead compound is the first step in making a new drug to treat diseases,and we identified, by in vitro experiments, molecules that inhibit the SARS-CoV-2 3CLprotease.There is space for improving by chemical modifications. The development processes take time,and we are presenting here the effect of both molecules in enzymatic assays and looking for the action of a repurposed drugs against the replication process of Sars-CoV-2.

Comment2

Since suramin and quinacrine target specific amino acids on the SARS-CoV-2 3CLpro, one concern that may arise is mutations that cause SARS-CoV-2 to escape their effects. In this manuscript, there is no mention of the specific strain that was used to perform these experiments. Do all strains possess these same amino acids at 3CLpro? Please elaborate on which strain that was used. In addition, if these amino acids differ, to strengthen the argument please elaborate on the potential efficacy of Suramin and Quinacrine on different SARS-CoV-2 variants (ex. The UK variant B.1.1.7, The South African variant, or other variants that may arise in the future).

Response 2 3CLproDNA was used based on the hCoV-19/Wuhan/WIV04/2019 genome. We included the information in the material and methods section (2.1. Cloning, expression,and purification of SARS-CoV-2 3CLpro, line 108).The occurring SARS-CoV-2 variants (e.g.,B.1.1.7, B.1.351,and P.1) carry mutations in several genes, e.g.,Orf1, Orf3, Orf8, envelope protein, N-terminus of the polyprotein, but especially in the spike glycoprotein. In the identified variants, the 3CL protease is unaffected from the mutations and potential inhibitors targeting the protease could helpcombat the different variants. The conclusion section has been changed accordingly(Lines 605to 617). Summarized information can be found in Supplementary Table 4.

Minor corrections:

Comment3 The title should be changed to “The repurposed drugs suramin and quinacrine cooperatively inhibit SARS-CoV-2 3CLproin vitro

Response 3We changed the title accordingly.

Comment4

The use of articles (ex. ‘the an’) is lacking in some parts of the manuscript. Please proofread and add as necessary.

Response 4Thank you for the comment, we proofread the manuscript and corrected accordingly.

Comment5The sentence “Drug repositioning offers hope to the SARS-CoV-2 control” at lines 31-32 does not make sense. Please modify this sentence to improve the meaning such as “Drug repositioning offers hope to control the SARS-CoV-2 pandemic.”

Response 5We changed accordingly in the abstract lines34 to36.

Comment6

In line 71, the author states that there are currently no efficacious drugs and vaccines available to combat SARS-CoV-2 infection, however, there are now authorized vaccines for use in humans (ex. Moderna, Pfizer-BioNTech COVID-19 vaccine). This sentence should address these authorized vaccines, their potential drawbacks, and how drug treatments such as Suramin and Quinacrine can fill these disadvantages.

Response 6Thank you for the comment. Text is changed accordingly in the abstract, lines 21 to 23,and in the introduction section lines 75 to 80.

Comment7

At line 108, the sentence ‘which was previously extensively washed with the lysis buffer and was extensively washed with the same buffer’ is unclear. Please change this sentence to be grammatically correct and clearer to the reader.

Response 7

The sentence has been rewritten (Line 125).

Comment8Please change ‘has grown overnight’ to ‘was grown overnight’ at line 95.

Response 8We changed accordingly(line 112).

Comment9At lines 145-147, the authors state that chloroquine did not show a satisfactory inhibitory effect. Please elaborate on this by including the range of concentrations tested in this experiment.

Response 9Thank you for the comment. We changed the position of the sentence to lines 163and 164. For the initial screening,we have used 10 μM of each tested drug. Based on the preliminary results, we decide to exclude chloroquine from further investigations.

Comment10Please remove theextra period at line 192.

Response 10We changed accordingly.

Comment11Throughout the manuscript the use of numerical numbers below ten are used (ex. 1-10), however, in line 198 the word ‘four’ is used. Please make sure the numbers are consistent (ex. Change to 4).

Response 11We changed words to numbers in several lines(e.g. 233, 234, 245,331, 338, 362, 377, 582).

Comment12At line 275 “Which is in good agreements...” is not a complete sentence. Please improve this sentence by changing the wording or adding it to the previous sentence.

Response 12Thank you for the comment. The sentence has been rewritten (Results section 3.1, line 314).

Comment13From lines 334-337, the authors mention other repurposed drug molecules (boceprevir, menadione, and disulfiram).

Response 13Boceprevir is a protease inhibitor used to treat hepatitis caused by hepatitis C virus (HCV). Disulfiram is a drug used to support the treatment of chronic alcoholism, but has been studied as a possible treatment for cancer and latent HIV infection. Menadione is an intermediate in the chemical synthesis of vitamin K and allowed as a nutritional supplement and it is used in the treatment of hypopro-thrombinemia. We added this information and references in lines383 to 388. Additionally, we added in table 5 (Line 601) a further column (inhibition type), where we mention how the drugs inhibit the 3CL protease, competitive (binds in the active site or substrate binding site) or noncompetitive (binds in an allostericsite).

Comment14At line 485, change ‘approx.’ to approximately.

Response 14We changed accordingly(line 559).

Comment15At line 299, the link to access the supplementary materials does not work.

Response 15Thank you for the comment, but we have no control in the link to the Supplementary material. I hope it will be now active

Reviewer 2 Report

The manuscript entitled “The repurposed drugs suramin and quinacrine inhibitor cooperatively in vitro SARS-CoV-2 3CLpro” by Eberle et al. characterizes the inhibitory activity of two candidate small molecules of the SARS-CoV-2 Main protease (Mpro) in vitro.  Using purified recombinant Mpro, the authors characterized the mechanisms of enzymatic inhibition of suramin and quinacrine in vitro.  Subsequent molecular modeling of the inhibitors on publicly available structural information for Mpro provides potential further molecular insights into the mechanisms of inhibition.  The overall goal to repurpose existing FDA approved drugs to treat patients effected by COVID19 is highly relevant and important during this current pandemic, as few therapeutics have efficacy at various stages of COVID19.  Furthermore, the number of direct acting anti-virals against SARS-CoV-2 proteases remains underdeveloped at this time.

Overall, the manuscript is well written, requiring minor editorial changes.  The authors have nicely established a recombinant protein expression system for Mpro.  The overall interpretation of the in vitro enzymatic assay results is consistent with the high-quality data shown in the manuscript.   The major weakness of the manuscript is the scope of the work presented.  The further study of the anti-viral mechanisms of Suramin is justified, however, the activity of quinacrine appears to be restricted to Vero cells without further validation in in vitro human cell models.  This leaves this reviewer to question the significance of the results for quinacrine as an antiviral for SARS-CoV-2 without further cellular data.  Furthermore, the modeling is data is quite speculative and represents untested hypothesis rather than experimental data.  In the absence of additional data to confirm hypotheses generated from the molecular modeling, it is the opinion of this reviewer that the manuscript is insufficiently developed for publication at this time.

Additional comments:

  1. The introduction states that no clinically safe vaccine is available to combat SARS-CoV-2. At the time of this review, there are 3 vaccines with clinical safety and efficacy that have been approved for emergency use in humans around.  The authors should modify the text to reflect the change in available vaccines.
  2. In addition, remdesivir is approved for emergency use and shown some efficacy at minimizing hospitalization. Other drugs (e.g. dexamethasone) have showed some clinical benefit in certain stages of COVID19.  This reviewer does agree that the number of available treatments, either prophylactic or therapeutic, are limiting which requires additional efforts to identify novel strategies or candidates to try in combination with existing standard of care for hospitalized patients.
  3. Have the authors tested the candidate inhibitors for promiscuous inhibitory (e.g. aggregation based inhibition, redox cycling, etc.)   It is well documented that promiscuous inhibitors can possess dose-dependent inhibitory activity.
  4. The fluorescence spectroscopy experiments are difficult to interpret. Specifically, the significance of the small shifts is difficult to interpret without data from a positive control.  Furthermore, complimentary orthologous approaches such as ITC, SPR, interferometry, or crystallization (or others) would provide greater confidence in the direct binding data.
  5. This reviewer believes the manuscript would be strengthened by testing hypotheses from the molecular modeling.  Are the amino acid residues at the putative sites of inhibitor binding required for enzymatic activity?  Do mutations effect enzymatic activity and/or inhibitor functionality.  Is it possible to crystalize suramin with Mpro to validate the docking hypotheses?

Author Response

Comment1

Overall, the manuscript is well written, requiring minor editorial changes. The authors have nicely established a recombinant protein expression system for Mpro. The overall interpretation of the in vitro enzymatic assay results is consistent with the high-quality data shown in the manuscript. The major weakness of the manuscript is the scope of the work presented. The further study of the antiviral mechanisms of Suramin is justified, however, the activity of Quinacrine appears to be restricted to Vero cells without further validation in in vitro human cell models. This leaves this reviewer to Comment the significance of the results for Quinacrine as an antiviral for SARS-CoV-2 without further cellular data. Furthermore, the modeling is data is quite speculative and represents untested hypothesis rather than experimental data. In the absence of additional data to confirm hypotheses generated from the molecular modeling, it is the opinion of this reviewer that the manuscript is insufficiently developed for publication at this time.

Response 1There is already published data for the antiviral effect of Quinacrine in human cell models using human pluripotent stem cells,and human pluripotent stem cells derived lung organoids (Han, Yuling, et al. "Identification of SARS-CoV-2 inhibitors using lung and colonic organoids." Nature 589.7841 (2021): 270-275.DOI: 10.1038/s41586-020-2901-9). This study demonstrates the potential of Quinacrine as an antiviral compound targeting SARS-CoV-2. The MD simulation data is in agreement with our experimental data. We identified Quinacrine as a competitive inhibitor and Suramin as a noncompetitive inhibitor. As a competitive inhibitor, Quinacrine binds in the active or the substrate-binding sites and competes with the substrate. Our MD simulations demonstrate that the Quinacrine/3CLprocomplex is stableover 200 ns. The same can be observed for Suramin. The MD simulations were performed for a total period of over200 ns(details in the section “simulation setup”), which is longer than other published simulations(examples, see below):200 ns https://doi.org/10.1038/s41598-019-42935-y150 ns https://doi.org/10.1080/07391102.2015.1046934https://doi.org/10.1016/j.jmgm.2017.03.002100 nshttps://doi.org/10.1016/j.lfs.2020.11808050 nshttps://doi.org/10.1080/07391102.2020.1808077

Additional comments:

Comment2

The introduction states that no clinically safe vaccine is available to combat SARS-CoV-2. At the time of this review, there are 3 vaccines with clinical safety and efficacy that have been approved for emergency use in humans around. The authors should modify the text to reflect the change in available vaccines.

In addition, remdesivir is approved for emergency use and shown some efficacy at minimizing hospitalization. Other drugs (e.g. dexamethasone) have showed some clinical benefit in certain stages of COVID19. This reviewer does agree that the number of available treatments, either prophylactic or therapeutic, are limiting which requires additional efforts to identify novel strategies or candidates to try in combination with existing standard of care for hospitalized patients.

Response 2 Thank you for the comment. We actualized the abstract, lines 21 to 24,and the introduction lines 75 to 80.

Comment4 Have the authors tested the candidate inhibitors for promiscuous inhibitory (e.g. aggregation-based inhibition, redox cycling, etc.). It is well documented that promiscuous inhibitors can possess dose-dependent inhibitory activity.

Response 4 To exclude inhibitors possibly acting as aggregators, a detergent-based control was performed by adding 0.001% freshly made up Triton X-100 to the reaction at the same time. If a molecule would exhibit significant inhibition of 3CLpro, which is diminished by detergent, it is almost certainly acting as an aggregation-based inhibitor (Feng et al. 2006, details see below), which was not observed for Quinacrine and Suramin.We added a related statement in the material/methods (Lines 170 to 172) and results (Lines 351to 355).Feng, B. Y., & Shoichet, B. K. (2006). A detergent-based assay for the detection of promiscuous inhibitors. Nature protocols, 1(2), 550-553.Comment5The fluorescence spectroscopy experiments are difficult to interpret. Specifically, the significance of the small shifts is difficult to interpret without data from a positive control. Furthermore, complimentary orthologous approaches such as ITC, SPR, interferometry, or crystallization (or others) would provide greater confidence in the direct binding data.Response 5We performed additionalstudies usingSPR. The experiments were performed as duplicates,and the results are shown as mean ± STD. In the section Material and Methods,we include information related to SPR experiments (2.5. Determination of dissociation constant using surface plasmon resonance)(Lines 191to 219) and the Results section (3.4 Investigation of SARS-CoV-2 3CLprointeraction with Suramin and Quinacrine using fluorescence spectroscopy and SPR) have been changed accordingly (Lines 391to 431). Additionally,figures and tables have been included in Supplementary material (S5, S6,S7 and Table S1).

The KD determination based on the fluorescence experiments was removed from the manuscript. The fluorescence spectroscopy titration experiments demonstrate the red edge excitation shift induced by the binding of suramin. Comment6This reviewer believes the manuscript would be strengthened by testing hypotheses from the molecular modeling. Are the amino acid residues at the putative sites of inhibitor binding required for enzymatic activity? Do mutations effect enzymatic activity and/or inhibitor functionality. Is it possible to crystalize Suramin with Mpro to validate the docking hypotheses?Response 6The MD simulation is in agreement with our experimental data. We identified Quinacrine as a competitive inhibitor and Suramin as a noncompetitive inhibitor, as discribed in the manuscript. As Quinacrine possesses a competitive inhibition, Quinacrinebinds in the active site or substrate binding site and competes with the substrate. The MD simulation demonstrates a stable complex over 300 ns. The same can be observed for Suramin in a possible allosteric site.As already mentioned in “response 1”, the MD simulations were performed for a total period of over 200 ns (details in the section “simulation setup”), which is longer than other published simulations (examples, see below):200 ns https://doi.org/10.1038/s41598-019-42935-y150 ns https://doi.org/10.1080/07391102.2015.1046934https://doi.org/10.1016/j.jmgm.2017.03.002100 nshttps://doi.org/10.1016/j.lfs.2020.11808050 nshttps://doi.org/10.1080/07391102.2020.1808077By the way, our scientific workwas the identification of inhibitor molecules targeting SARS-CoV-2 3CLprousing repurposed drugs that could inhibit the protease and work as a lead molecule. By inhibition assays, inhibition mode,and in silico binding interaction studies,we characterized and described the inhibition interaction by Quinacrine,and Suramin. The binding region of Suramin was suggested using docking with subsequent MD simulations. We make clear in the manuscript that this binding region in the protease has the potential to bind to ligands, but of course,structural methods are necessary to confirm that suggestion.Co-crystallization experiments of SARS-CoV-2 3CLprowith Quinacrine, Suramin,and Quinacrine + Suramin are planned.

Reviewer 3 Report

The study of Eberle et al. attempts to provide a molecular framework to previous data that showed that suramin and quinacrine inhibit SARS-CoV-2 replication. In the current study the authors point to the viral protease as the drug target for the two compounds. Moreover, they conduct experimental inhibition and binding studies that show that both drugs inhibit the protease with an IC50 of a few micro molar. These results are consistent with the in vitro data of the compounds. Moreover, the authors present convincing data that show that while one of the compounds inhibits competitively, the other does not. With this idea in mind, the authors speculated that the two drugs might work cooperatively, which indeed they do.

On the whole this is a sound and interesting study that provides a molecular picture of an anti viral agent. I have little to critique the study nor the conclusions expect for the following few minor points:

1. The third sentence in the abstract states that: "To date, no clinically safe drug or vaccine is available and the development of molecules to combat SARS-CoV-2 infections is imminent."  This is obviously wrong. There are several vaccines and anti COVID-19 agents (dexamethasone for example) that are approved. How do the authors not know this???

2. The manuscript should be proof read in order to correct numerous English errors.

Author Response

Comment1

The third sentence in the abstract states that: "To date, no clinically safe drug or vaccine is available and the development of molecules to combat SARS-CoV-2 infections is imminent." This is obviously wrong. There are several vaccines and anti COVID-19 agents (dexamethasone for example) that are approved. How do the authors not know this???

Response 1 Thank you for the comment, this manuscript was uploaded on a preprint server(November 2020), and unfortunately,we have submitted the old version to Viruses.

The manuscript was updated accordingly.Abstract:lines 21 to 24 and Introduction section:lines 75 to 80.Comment2The manuscript should be proof read in order to correct numerous English errors.

Response 2The manuscript has been revised.

Reviewer 4 Report

The study report that two repurposed drugs quinacrine and suramin could inhibit SARS-CoV-2 3CLpro activity competitive and non-competitive, respectively. Docking and molecular dynamics simulations were used to identify the possible binding mode of the two drugs with 3CLpro. And suramin in combination with quinacrine showed promising synergistic efficacy to inhibit SARS-CoV-2 3CLpro. Some recent study show that suramin may target SARS2 RdRp; and quinacrine may target ACE2 and furin (Detailed below in comments against Line305). In this study, it’s interesting that authors here find that the two drugs target a different SARS2 protein, namely 3CLpro, which is also an appealing anti-CoVs target. However, I do have some concerns that needed to be addressed.

Recommendations for the authors

Major concerns

Point 1 The authors used fluorescence spectroscopy to measure the binding (Kd) between drugs and 3CLpro. It should be better to use more than one method or any other more direct technology (like SPR,) to measure the binding between the drugs and 3CLpro, since it’s the most important and the foothold of this study.

Point 2 The authors used docking and molecular dynamic simulations to predict the possible binding mode of the two drugs against 3CLpro. There is no doubt this method is very practical and give some useful information when it’s difficult to obtain the crystal structure of the complex. However, mutagenesis study will make the conclusion more convincing. For example, quinacrine bind Met165 and Gln189 through H-bond, and bind His41through hydrophobic interaction. Thus, the 3CLpro mutants M165A, Q189A and H41A should have negative impacts on the bindings. As well as the double site mutant and multiple site mutant. And when selecting the sites for mutation, it’s better to make sure the mutations will not result in big structure changes based on the apo-3CLpro structure.

Minor points

Line 21 “To date, no clinically safe drug or vaccine is available” seems not accurate. Some vaccines have already been approved, like the Pfizer-BioNTech COVID-19 Vaccine.

Line 60 some coronaviruses do not have nsp1, so they will have 15 NSPs.              

Line 62 Authors should indicate that these NSPs will also generate the genomic RNA.

Line 93 The construct contains an N-terminal GST-tag and a PreScission protease cleavage site (LEFLFQGP). After cleavage, will any residues remain in the N-terminus of the 3CLpro? 3CLpro N-terminus is located between the two protomers, so will any remaining N-terminal residues affect the enzymatic activity?

Line 305 “however, the target protein of both molecules in the viral replication process was not identified.” This may not be accurate. Some of the target proteins were also reported.

Quinacrine dihydrochloride (QNHC) was reported to bind with ACE2 (SARS-CoV-2 receptor), and treatment with QNHC could also decrease the expression levels of furin (furin is responsible for the cleavage of the SARS-CoV-2 spike between S1 and S2, thus important for viral cell entry). See “Yuling Han et al., Identification of SARS-CoV-2 inhibitors using lung and colonic organoids, 2020, nature”.

Suramin was reported to bind to SARS-CoV-2 nsp12 (RdRp), acting as a potent inhibitor of the RdRp through blocking the binding of RNA to the enzyme. See “Wanchao Yin et al., Structural basis for repurposing a 100-years-old drug suramin for treating COVID-19. Preprint at research square and bioRxiv.” This statement should be revised, and citation should be updated.

Line 486 The authors assume that the conformational changes after suramin interaction with SARS-CoV-2 3CLpro induce a structural change in both active sites in the dimer, which could make them more accessible for the interaction with quinacrine. Will it be possible to dock the quinacrine molecular into the active sites of the 3CLpro after suramin have been docked into the 3CLpro? Then analyze whether the structural changes in both active sites will result in a better binding with quinacrine.

Author Response

Comment1

The authors used fluorescence spectroscopy to measure the binding (Kd) between drugs and 3CLpro. It should be better to use more than one method or any other more direct technology (like SPR,) to measure the binding between the drugs and 3CLpro, since it’s the most important and the foothold of this study.Response 1We performed additional experiments using SPRto study the interaction between the drugs and 3CLpro. The experiments were performed as duplicates and we included information related to the methodology in section Material and methods, subsection 2.5. Determination of dissociation constant using Surface Plasmon Resonance(Lines 191to 219) and the results were included in subsection 3.4 Investigation of SARS-CoV-2 3CLprointeraction with Suramin and Quinacrine using fluorescence spectroscopy and SPR (Lines 391to 431), additionally three supplementary figures and a supplementary table were added (Fig.S5, S6,S7 and Table S1).The KD determination based on the fluorescence experiments was removed from the manuscript. The fluorescence spectroscopy titration experiments demonstrate the red edge excitation shift induced by the binding of suramin.

Comment2

Point 2 The authors used docking and molecular dynamic simulations to predict the possible binding mode of the two drugs against 3CLpro. There is no doubt this method is very practical and gives some useful information when it’s difficult to obtain the crystal structure of the complex. However, mutagenesis study will make the conclusion more convincing. For example, quinacrine bind Met165 and Gln189 through H-bond, and bind His41through hydrophobic interaction. Thus, the 3CLpromutants M165A, Q189A and H41A should have negative impacts on the bindings. As well as the double site mutant and multiple site mutant. And when selecting the sites for mutation, it’s better to make sure the mutations will not result in big structure changes based on the apo-3CLprostructure.Response 2Thank you for your comment. Site direct mutagenesis studies will confirm the binding moiety.However,in the first phase of our studies, we would like to identify inhibitor molecules using repurposed drugs that would be able to inhibit the protease and work as a lead molecule. By inhibition assays, inhibition mode,and in silico binding interaction studies,we were able to characterize and describe the inhibition mode of the two drugs Quinacrine and Suramin. At the moment,we are setting up co-crystallization studies.Using MD simulations, we let clear in the manuscript that the binding regions we are showing have the potential to bind ligands and structural methods are necessary to confirm that suggestion.

Minor points

Comment3Line 21 “To date, no clinically safe drug or vaccine is available” seems not accurate. Some vaccines have already been approved, like the Pfizer-BioNTech COVID-19 Vaccine.Response 3Thank you for the comment. We have updated the manuscript: Abstract (lines 21 to 24)and Introduction (lines 75 to 80).Comment4Line 60 some coronaviruses do not have nsp1, so they will have 15 NSPs.Response 4Thank you, the sentence have been rewritten(Lines64 and65).Comment5Line 62 Authors should indicate that these NSPs will also generate the genomic RNA.Response 5The sentence have been rewritten(Line 66).Comment6Line 93 The construct contains an N-terminal GST-tag and a PreScission protease cleavage site (LEFLFQGP). After cleavage, will any residues remain in the N-terminus of the 3CLpro? 3CLproN-terminus is located between the two protomers, so will any remaining N-terminal residues affect the enzymatic activity?

Response 6 The PreScissionprotease cleaves between Q and G (LEFLFQGP). Two residues, GP, will remain on the N-terminus of 3CLpro. In section 3.2 we describe the Michaelis Menten kinetics of the protease and compared the Kcat/KMvalue with reported values in the literature. Our value is comparable to theone reported by Zhanget al. 2020 (Crystal structure of SARS-CoV-2 main protease provides a basis for the design of improved α-ketoamide inhibitors.Science), which demonstrates that the extra two residues (GP) do not affect the activity of the protein.Comment7Line 305 “however, the target protein of both molecules in the viral replication process was not identified.” This may not be accurate. Some of the target proteins were also reported.

Quinacrine dihydrochloride(QNHC) was reported to bind with ACE2 (SARS-CoV-2 receptor), and treatment with QNHC could also decrease the expression levels of furin (furin is responsible for the cleavage of the SARS-CoV-2 spike between S1 and S2, thus important for viral cell entry). See “Yuling Han et al., Identification of SARS-CoV-2 inhibitors using lung and colonic organoids, 2020,nature”.Suramin was reported to bind to SARS-CoV-2 nsp12 (RdRp), acting as a potent inhibitor of the RdRp through blocking the binding of RNA to the enzyme. See“Wanchao Yin et al., Structural basis for repurposing a 100-years-old drug suramin for treating COVID-19.Preprint at research square and bioRxiv.” This statement should be revised, and citation should be updated.Response 7Thank you for the comment;we included the missing information and references accordingly. Section Results, subsection:3.3Inhibition assay of chloroquine, Quinacrine,and Suramin against SARS-CoV-2 3CLpro, lines344to 347.Comment8Line 486 The authors assume that the conformational changes after suramin interaction with SARS-CoV-2 3CLproinduce a structural change in both active sites in the dimer, which could make them more accessible for the interaction with Quinacrine. Will it be possible to dock the Quinacrine molecular into the active sites of the 3CLproafter Suramin have been docked into the 3CLpro? Then analyze whether the structural changes in both active sites will result in a better binding with Quinacrine.Response 8We performed docking and MD simulations (for 200 ns) of Quinacrine and used the SARS-CoV-2 3CLpro-suramin complex after MD simulations as starting structure. One of the Quinacrine leaves the active site because it collapsed, as already described in figure 7. The position of Quinacrine in the second active site changes but block still the His41 residue. Two hydrogen bonds stabilize the interaction with Quinacrine(the involved amino acids are shown in Figure 8 and Table S3). In addition, the position of suramin changes and one residue of the second protein monomer forms anH-bond, which didn ́t appear before. A comparison between 3CLpro-quinacrine and 3CLpro-suramin + Quinacrine demonstrated changes in the active site's secondary structure, which are presented in detail in Figure 8. We corrected also the sentence that both active sites would be more accessible,because one active site collapsed. This new MD simulation is described in section Results, subsection3.6Suramin and Quinacrine act cooperatively to inhibit SARS-CoV-2 3CLpro(Lines 560to 575),Figure 8, Supplementary Fig. 14 and Supplementary Table 3.

Round 2

Reviewer 2 Report

The revised manuscript entitled “The repurposed drugs suramin and quinacrine cooperatively inhibit SARS-CoV-2 3CLpro in vitro” by Eberle et al. characterizes the inhibitory activity of two candidate small molecule drugs of the SARS-CoV-2 Main protease (Mpro) in vitro.  Using purified recombinant Mpro, the authors characterized the mechanisms of enzymatic inhibition of suramin (noncompetitive) and quinacrine (competitive) in vitro.  The addition of Surface Plasmon Resonance (SPR) data to characterize the binding affinity of these two drugs to SARS-CoV2 is added value to the biochemistry in this manuscript.  The thorough and detailed molecular modeling of the inhibitors on publicly available structural information for Mpro is very consistent with the biochemical data and provides further molecular insights into the possible mechanisms of inhibition of the protease, which may broadly facilitate development of direct acting antivirals against SARS-CoV2.  The goal to repurpose existing FDA approved drugs to treat patients effected by COVID19 is highly relevant and important during this current pandemic, as few therapeutics have efficacy against COVID19.  Especially drugs that are inexpensive and can be easily distributed in developing countries.  Furthermore, the most effective anti-virals to date in the clinic are those that directly act on viral targets (e.g. HIV, HCV), and therefore, effort to better define direct acting anti-virals against SARS-CoV2 remains underdeveloped at this time and an important area of study.

Overall, the revised manuscript is well written, requiring a few minor editorial changes (outlined below).  The strength of the original and the revised manuscript is the biochemistry.  The interpretation of the in vitro enzymatic assay results is consistent with the high-quality data shown in the manuscript.   The addition of SPR data further strengthens the data to quantitatively characterize the binding of quinacrine and Suramin to Mpro.  Although the molecular modeling is consistent with the biochemical data, the authors state that these molecular details represent possible/theoretical conditions, the value of which is difficult to assess without additional validation experiments or structure. It is possible that quinacrine binds at a site other than the active site and still functions as a competitive inhibitor and the binding site of suramin is a hypothesis. This reviewer understand that the inclusion of structure changes the scope of the current manuscript and where it is potentially submitted for review and publication.  The additive/synergistic effect of the suramin/quinacrine combination in vitro is intriguing and the anti-viral activity of the drugs as single agents and as a combination warrants further investigation in human cell culture models to validate the biochemical data.

It is the opinion of this reviewer that the modeling requires additional biochemical validation and that the anti-viral activity of these inhibitors be further confirmed.

Additional comments:

  1. There are several instances in the manuscript were paragraphs have redundant sentences or wording that can be edited to be more concise. For example:  lines 413 to 420
  2. Reference manger a several sites enters two separate reference markers, versus one concise reference marker. This is a simple editorial fix. For example: line 403 and 405.
  3. The authors have stated that assays including low concentrations (0.001%) of non-ionic detergents have no impact on the inhibitory activity of quinacrine and suramin. It is not clear if the data for this experiment is shown.  The Feng et. al. referenced in the revised manuscript suggests 0.01% and references therein indicate that higher concentrations of non-ionic detergents (upto 0.1%) may be required to disrupt the aggregation-based inhibition.  A detergent titration plus/minus inhibitors would more fully demonstrate the impact of detergent on inhibition by quinacrine/suramin versus disruption of the enzymatic activity by detergent.  Furthermore, the assay buffer contains the strong reducing agent TCEP.  Promiscuous inhibition by redox-cycling and production of H2O2 can demonstrate dose response inhibition (Johnston et. al. Assay and Drug development Technologies, 2010).  There is no data presented to rule out this type of promiscuous inhibition.
  4. For SPR data, it typical to report K on and K off rates in addition to the apparent KD (Koff/Kon).  Table 2 can easily be modified to include these data.
  5. There are many reports of anti-viral activity for candidate drugs in a variety of cell models. There is also poor validation of anti-viral activity of drugs across cellular models and laboratories.  The manuscript would be strengthened by including anti-viral experiments that validate the activity of suramin and quinacrine and the combination in in vitro human cell models.  The reviewer understand that these experiments are not standard to all laboratories due to the need for BSL3 access.  It is also the opinion of this reviewer that anti-viral assays are a requirement for prioritizing candidates for further follow-up.

Reviewer 4 Report

The authors addressed most of my questions, and the manuscript is indeed improved.

Minor points

Line 350 “3CL protease” should be kept same with 3CLpro throughout the manuscript. And the same with “3CL protease” in line 616 and “3CLpro” in line 638.

Line 615 “Sars-CoV-2” should be changed to “SARS-CoV-2”, so keep it same throughout the manuscript.

Figure 6 SARS-CoV2 should be changed to SARS-CoV-2
